# Combining Analytical Smoothing with Surrogate Losses for Improved Decision-Focused Learning

## Abstract

Many combinatorial optimization problems (COPs) in routing, scheduling, and assignment involve parameters such as price or travel time that must be predicted from data; so-called predict-then-optimize (PtO) problems. Decision-focused learning (DFL) is a family of successful end-to-end techniques for PtO that trains machine learning models to minimize the error of the downstream optimization problems. This requires solving the COP for each training instance with the predicted parameters and computing the derivative of the solution with respect to the predicted parameters—tasks that become computationally prohibitive for large COPs. When the COP is an integer linear program (ILP), a recent work, DYS-Net, applies Davis-Yin splitting (DYS) to solve and differentiate through quadratically regularized ILP. While this fully neural approach significantly accelerates training, it has only been evaluated on datasets where true cost parameters are unobserved, limiting its comparability to state-of-the-art techniques. In this work, we experimentally demonstrate that minimizing empirical regret using DYS-Net results in suboptimal regret on test data compared to state-of-the-art DFL methods across three different COPs. We attribute this to the plateau effect: regret remains constant over regions of the parameter space, with sharp changes occurring only at transition points resulting in low gradient values over much of the space when regret is minimized. We illustrate how minimizing a noise contrastive surrogate loss avoids this problem. Through extensive experiments, we show that minimizing this surrogate loss allows DYS-Net to achieve test regret levels that are comparable to or lower than the state-of-the-art methods. Moreover, by achieving state-of-the-art regret levels with significantly reduced training times, our approach represents a substantial advance in DFL research, particularly in improving its scalability towards large-scale PtO problems.

## 1 Introduction

Many decision-making problems in real-world can be cast as optimization problems. Some parameters of these optimization problems are often unknown due to uncertainty or the anticipation of future events. As prediction of these parameters is crucial for making high-quality decisions, leveraging contextual information is important at prediction time. The availability of historical data, combined with the rapid growth of predictive machine learning (ML), has fueled increasing interest in data-driven contextual optimization (Sadana et al., 2025).

When the goal is to predict parameters (such as cost or travel time) of an optimization problem, such problems can be viewed as "predict-then-optimize"(PtO) problems, including two key steps— the prediction of the unknown parameters and the subsequent optimization using those predicted parameters. Prediction-focused learning is the approach to tackle PtO problems by treating the prediction step independent of the optimization step, based on the assumption that increasing accuracy of predictions would lead to good quality decisions. However, in practice, ML models fail to achieve 100% accuracy, and in the presence of prediction errors, such a prediction-focused approach fails to consider how the error in predictions impacts the solution to the optimization problem. This fact motivates the research in decision-focused learning (DFL), as surveyed by Mandi et al. (2024).

DFL trains ML models to predict the uncertain parameters by *directly* minimizing the task loss, which reflects the quality of the solutions made using the predicted parameters. Gradient-based DFL entails computing the derivative of the optimization problem's solution with respect to the predicted parameters. However, for combinatorial optimization problems (COPs), this derivative is almost always zero because slight parameter changes typically do not alter the solution, except at certain transition points where the derivative does not exist.

In this paper, we focus on predicting parameters of COPs, where the predicted parameters appear linearly in the objective function. Previous works in DFL use two broad categories of approaches for such problems: (a) turning the COP into a differentiable mapping by smoothing the optimization to a convex optimization problem (Wilder et al., 2019; Mandi & Guns, 2020), and then minimizing the task loss, and (b) using surrogate loss functions (Elmachtoub & Grigas, 2022; Mulamba et al., 2021; Mandi et al., 2022), for which gradients or subgradients exist. Both approaches require solving the (smoothed) COP for each training instance with the predicted cost parameter and computing the derivative of the COP solution with respect to the predicted parameter. This poses significant scalability challenges, especially for large-scale COPs.

To improve the scalability of DFL, McKenzie et al. (2024) recently developed a fast, fully differentiable neural optimization layer, DYS-Net, for integer linear programs (ILPs). DYS-Net can be viewed within the first category of DFL approaches, as it incorporates neural smoothing of the LP. As DYS-Net can be implemented entirely as a neural network, it significantly accelerates training. However, McKenzie et al. (2024) consider datasets where true cost parameters are unobserved, whereas in most PtO benchmark problems, as in (Tang & Khalil, 2023), it is assumed that the true parameters are observed. Consequently, DYS-Net has not been compared to state-of-the-art DFL techniques.

In this work, we consider the task of minimizing the empirical regret of the training set using DYS-Net in datasets, where the true cost parameters are observed. We argue that this would result in low gradient values over much of the space as regret remains constant over regions of the parameter space, with sharp changes occurring only at transition points. To address this, we propose to minimize the surrogate losses, even though it is possible to minimize regret directly by differentiating through the smoothed optimization problem. We justify the advantage of using a surrogate loss by comparing the pattern of the gradient landscape with respect to regret and the surrogate loss. In this way, this paper combines the two approaches of DFL. Minimizing the surrogate losses using DYS-Net allows us to accelerate DFL with regret lower than or equal to state-of-the-art DFL techniques.

In summary, this paper makes the following contributions:

- We show that although 'smoothing' makes the optimization problem differentiable, the gradient remains zero over most of the parameter space due to the plateau effect.
- To address the plateau effect which occurs even after smoothing, we combine the two families of DFL approaches by minimizing a surrogate loss post-smoothing.
- We empirically demonstrate that for smoothing approaches, minimizing surrogate losses results in lower regret on test data than minimizing the regret.
- We show that by minimizing the surrogate loss using DYS-Net achieves regret comparable to existing state-of-the-art methods while reducing training time by up to five-fold.

## 2 PREDICT-THEN-OPTIMIZE PROBLEM DESCRIPTION

In PtO problems, decisions are made by solving COPs. In this work, we focus on COPs with linear objectives and the prediction of objective function parameters. These COPs can be formulated as LPs or integer LPs (ILPs), both of which have extensive practical applications. Any LP can be transformed in the following standard LP form:

$$v^\star(y) = \arg\min_{v} y^\top v \quad \text{s.t.} \ Av = \mathbf{b}; \ v \geq \mathbf{0} \tag{1}$$

where $v \in \mathbb{R}^K$ is a decision variable and $v^\star(y)$ is the optimal solution for a given cost parameter $y \in \mathbb{R}^K$. ILPs differ from LPs in that the decision variables $v$ are restricted to integer values. For brevity, we use $\mathcal{F}$ to denote the feasible space. So, for the standard LP formulation, $\mathcal{F} = \{v \in \mathbb{R}^K | Av = \mathbf{b} \, ; v \geq \mathbf{0}\}$. Unless it is explicitly stated otherwise, $v^\star$ will denote $v^\star(y)$.

To account for uncertainty in the decision-making, PtO problems comprise two steps—the prediction of the unknown parameters and solving the optimization problem using the predicted parameters. We consider PtO formulation, where the vector of cost parameters $\boldsymbol{y}$ is not known prior to solving. Instead, a list of contextual information $\boldsymbol{\phi}$, correlated with $\boldsymbol{y}$ is available for predicting $\boldsymbol{y}$. In PtO problems, an ML model $\mathcal{M}_\omega$ (with trainable parameters $\omega$) is trained to map $\boldsymbol{\phi} \to \boldsymbol{y}$ using past observation pairs $\{(\boldsymbol{\phi}_i, \boldsymbol{y}_i)\}_{i=1}^N$. Given their success in predictive tasks, neural networks have become the preferred choice for the predictive modeling task in PtO problems.

A straightforward approach to the PtO problem is to train $\mathcal{M}_\omega$ to generate accurate parameter predictions $\hat{\boldsymbol{y}} = \mathcal{M}_\omega(\boldsymbol{\phi})$ by minimizing the prediction errors with respect to ground-truth $\boldsymbol{y}$. Previous works (Wilder et al., 2019; Elmachtoub & Grigas, 2022; Mandi et al., 2020) justify why such a *prediction-focused approach* produces suboptimal performance. By contrast, in *decision-focused learning* (DFL), the ML model is directly trained to optimize the task loss, the quality of the resulting decisions. When only the parameters in the objective function are predicted, the task loss of interest is typically *regret*, which measures the suboptimality of a decision resulting from prediction errors. The regret for making the decisions $\boldsymbol{v}$ under the true realization $\boldsymbol{y}$ can be expressed in the following form:

$$Regret(\boldsymbol{v}, \boldsymbol{y}) = \boldsymbol{y}^\top \boldsymbol{v} - \boldsymbol{y}^\top \boldsymbol{v}^\star(\boldsymbol{y}) \tag{2}$$

In PtO problems, one can consider other task losses (Appendix H), such as squared decision errors (SqDE) between $\boldsymbol{v}^\star(\boldsymbol{y})$ and $\boldsymbol{v}^\star(\hat{\boldsymbol{y}})$, i.e., $SqDE = ||\boldsymbol{v}^\star(\boldsymbol{y}) - \boldsymbol{v}^\star(\hat{\boldsymbol{y}})||^2$.

## 3 DECISION-FOCUSED LEARNING FOR COMBINATORIAL OPTIMIZATION

The DFL approach trains $\mathcal{M}_\omega$ to directly minimize $\frac{1}{N} \sum_{i=1}^N Regret(\boldsymbol{v}^*(\mathcal{M}_\omega(\boldsymbol{\phi}_i)), \boldsymbol{y}_i)$, the empirical risk minimization counterpart of $\mathbb{E}[Regret(\boldsymbol{v}^*(\mathcal{M}_\omega(\boldsymbol{\phi})), \boldsymbol{y})]$ since the true distribution is unknown. This minimization of regret in gradient descent-based learning requires backpropagation through the COP, which involves computing the derivative of $\boldsymbol{v}^\star(\hat{\boldsymbol{y}})$ with respect to $\hat{\boldsymbol{y}} = \mathcal{M}_\omega(\boldsymbol{\phi})$. While $\frac{d\boldsymbol{v}^\star(\hat{\boldsymbol{y}})}{d\hat{\boldsymbol{y}}}$ can be computed for convex optimization problems through implicit differentiation (Agrawal et al., 2019; Amos & Kolter, 2017), it is more challenging when the optimization problem is combinatorial. This is because when the parameters of a COP change, the solution either remains unchanged or shifts abruptly, meaning the derivatives are almost always zero and undefined at abrupt changes.

Broadly there are two approaches of implementing DFL for COPs: (a) smoothing the COP to a smooth convex optimization problem and (b) using a surrogate loss that is differentiable. For a detailed discussion on how existing DFL techniques tackle this challenge, we refer readers to the survey paper by Mandi et al. (2024).

### 3.1 DIFFERENTIABLE OPTIMIZATION BY SMOOTHING OF COMBINATORIAL OPTIMIZATION

To address this, methodologies in this category modify the optimization problem by smoothing and then analytically differentiate the 'smoothed' optimization problem. (Readers may refer to Appendix I for detailed explanation.) In this work, we focus on optimization problems with linear objective functions such as LPs and ILPs. For LPs, Wilder et al. (2019) propose transforming the LPs into 'smoothed' quadratic programs (QPs) by augmenting the objective function with the square of the Euclidean norm of the decision variables in the following form:

$$\min_{\boldsymbol{v}} \hat{\boldsymbol{y}}^\top \boldsymbol{v} + \mu \|\boldsymbol{v}\|_2^2 \ \text{ s.t. } \ A\boldsymbol{v} = \mathbf{b} \ ; \boldsymbol{v} \geq \mathbf{0} \tag{3}$$

where $\mu \geq 0$ is the smoothing parameter, controlling the strength of smoothing. After smoothing, the solution is not restricted to being at a vertex of the LP polyhedron. In the 'smoothed' problem, unlike the original LP, the solution do not change abruptly. The solution either may not change or change smoothly with the change of the cost vector, as illustrated in Figure 1b. Consequently, $\boldsymbol{v}^\star(\hat{\boldsymbol{y}})$ becomes differentiable with respect to $\hat{\boldsymbol{y}}$. The QP smoothing approach has been applied in various DFL works (Ferber et al., 2020; 2023; McKenzie et al., 2024). Mandi & Guns (2020) consider another form of smoothing by adding logarithm barrier term into the LP. When the underlying optimization problem is an ILP, smoothing of the LP, resulting from the continuous relaxation of the ILP is carried out.

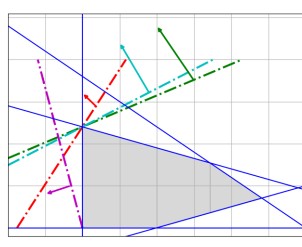
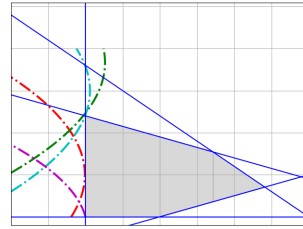

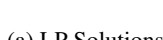

(a) LP Solutions          (b) Solutions to the QP smoothing

Figure 1: Schematic diagram showing the effect of QP smoothing. (a) LP solutions and the corresponding isocost line for four cost vectors. The green, cyan and red cost vectors result in the same solution, the top vertex, highlighting that a slight rotation of the isocost lines may not alter the LP solution. However, if the isocost lines rotate too much, for example, the violet line, the solution suddenly shifts to a different vertex. (b) The isocost lines change after applying QP smoothing, and the solution is no longer restricted to a vertex. For example, the red vector results in a smooth change in the solution. However, even with smoothing, some cost vectors, like the cyan and green, may still share the same solution.

### 3.2 SURROGATE LOSSES FOR DFL

Surrogate loss functions are used for training in DFL because they are crafted to have non-zero (sub)gradients everywhere while also directly correlating with the task loss—as regret decreases, surrogate loss functions decrease as well. We focus on two surrogate losses, used widely in DFL.

#### 3.2.1 SMART PREDICT THEN OPTIMIZE(SPO)

The SPO+ loss (Elmachtoub & Grigas, 2022), a convex upper bound of $Regret(\boldsymbol{v}^\star(\hat{\boldsymbol{y}}), \boldsymbol{y})$, is one of the first and most widely used surrogate losses for linear objective optimization problems.

$$Regret(\boldsymbol{v}^\star(\hat{\boldsymbol{y}}), \boldsymbol{y}) = \boldsymbol{y}^\top \boldsymbol{v}^\star(\hat{\boldsymbol{y}}) - \boldsymbol{y}^\top \boldsymbol{v}^\star = \boldsymbol{y}^\top \boldsymbol{v}^\star(\hat{\boldsymbol{y}}) - 2\hat{\boldsymbol{y}}^\top \boldsymbol{v}^\star(\hat{\boldsymbol{y}}) + 2\hat{\boldsymbol{y}}^\top \boldsymbol{v}^\star(\hat{\boldsymbol{y}}) - \boldsymbol{y}^\top \boldsymbol{v}^\star$$

$$\leq \max_{\boldsymbol{v}' \in \mathcal{F}}\{\boldsymbol{y}^\top \boldsymbol{v}' - 2\hat{\boldsymbol{y}}^\top \boldsymbol{v}'\} + 2\hat{\boldsymbol{y}}^\top \boldsymbol{v}^\star(\hat{\boldsymbol{y}}) - \boldsymbol{y}^\top \boldsymbol{v}^\star \leq \underbrace{\max_{\boldsymbol{v}' \in \mathcal{F}}\{\boldsymbol{y}^\top \boldsymbol{v}' - 2\hat{\boldsymbol{y}}^\top \boldsymbol{v}'\} + 2\hat{\boldsymbol{y}}^\top \boldsymbol{v}^\star - \boldsymbol{y}^\top \boldsymbol{v}^\star}_{\mathcal{L}_{SPO+}(\boldsymbol{v}^\star(\hat{\boldsymbol{y}}), \boldsymbol{y})}$$

Instead of minimizing $Regret$, they propose to minimize this convex upperbound, which is called $\mathcal{L}_{SPO+}(\boldsymbol{v}^\star(\hat{\boldsymbol{y}}), \boldsymbol{y})$ loss. It can be expressed in the following form:

$$\mathcal{L}_{SPO+}(\boldsymbol{v}^\star(\hat{\boldsymbol{y}}), \boldsymbol{y}) = \max_{\boldsymbol{v}' \in \mathcal{F}}\{2\hat{\boldsymbol{y}}^\top \boldsymbol{v}^\star - \boldsymbol{y}^\top \boldsymbol{v}^\star - (2\hat{\boldsymbol{y}} - \boldsymbol{y})^\top \boldsymbol{v}'\} = (2\hat{\boldsymbol{y}} - \boldsymbol{y})^\top \boldsymbol{v}^\star - \min_{\boldsymbol{v}' \in \mathcal{F}}\{(2\hat{\boldsymbol{y}} - \boldsymbol{y})^\top \boldsymbol{v}'\}$$

$$= (2\hat{\boldsymbol{y}} - \boldsymbol{y})^\top \boldsymbol{v}^\star - (2\hat{\boldsymbol{y}} - \boldsymbol{y})^\top \boldsymbol{v}^\star(2\hat{\boldsymbol{y}} - \boldsymbol{y}) \qquad (4)$$

They also propose the following sub-gradient for gradient-based training using any solver of choice:

$$\nabla_{\mathcal{L}_{SPO+}} = 2(\boldsymbol{v}^\star - \boldsymbol{v}^\star(2\hat{\boldsymbol{y}} - \boldsymbol{y})) \qquad (5)$$

#### 3.2.2 CONTRASTIVE LOSS

Mulamba et al. (2021) propose a surrogate loss based on noise-contrastive estimation (NCE) (Gutmann & Hyvärinen, 2012). The loss is derived by considering the log-likelihood ratio between $\boldsymbol{v}^\star$ and other feasible points $\boldsymbol{v}'$. By maximizing this likelihood, they propose to minimize the following NCE loss:

$$\mathcal{L}_{NCE}(\boldsymbol{v}^\star(\hat{\boldsymbol{y}}), \boldsymbol{y}) = \max_{\boldsymbol{v}' \in \mathcal{F}}\{\hat{\boldsymbol{y}}^\top \boldsymbol{v}^\star - \hat{\boldsymbol{y}}^\top \boldsymbol{v}'\} = \hat{\boldsymbol{y}}^\top \boldsymbol{v}^\star - \min_{\boldsymbol{v}' \in \mathcal{F}}\{\hat{\boldsymbol{y}}^\top \boldsymbol{v}'\} = \hat{\boldsymbol{y}}^\top \boldsymbol{v}^\star - \hat{\boldsymbol{y}}^\top \boldsymbol{v}^\star(\hat{\boldsymbol{y}}) \qquad (6)$$

Note that $\mathcal{L}_{NCE}$ is similar to $\mathcal{L}_{SPO+}$, except that in $\mathcal{L}_{NCE}$, $2\hat{\boldsymbol{y}} - \boldsymbol{y}$ is replaced with $\hat{\boldsymbol{y}}$. This introduces a shortcoming in $\mathcal{L}_{NCE}$. The minimum of $\mathcal{L}_{NCE}$, which is zero, can be achieved either when $\boldsymbol{v}^\star(\hat{\boldsymbol{y}}) = \boldsymbol{v}^\star$ or by predicting $\hat{\boldsymbol{y}} = 0$. To prevent minimizing $\mathcal{L}_{NCE}$ by predicting $\hat{\boldsymbol{y}} = 0$, Mulamba et al. (2021) further modify $\mathcal{L}_{NCE}$ to derive the self-contrastive estimation (SCE) loss:

$$\mathcal{L}_{SCE}(\boldsymbol{v}^\star(\hat{\boldsymbol{y}}), \boldsymbol{y}) = (\hat{\boldsymbol{y}} - \boldsymbol{y})^\top \boldsymbol{v}^\star - (\hat{\boldsymbol{y}} - \boldsymbol{y})^\top \boldsymbol{v}^\star(\hat{\boldsymbol{y}}) = \hat{\boldsymbol{y}}^\top \boldsymbol{v}^\star - \hat{\boldsymbol{y}}^\top \boldsymbol{v}^\star(\hat{\boldsymbol{y}}) + \boldsymbol{y}^\top \boldsymbol{v}^\star(\hat{\boldsymbol{y}}) - \boldsymbol{y}^\top \boldsymbol{v}^\star(\boldsymbol{y}) \qquad (7)$$

**Proposition 1.** *$\mathcal{L}_{SCE}(\boldsymbol{v}^{\star}(\hat{\boldsymbol{y}}), \boldsymbol{y})$ has the following properties ( proof is given in Appendix A):*

1. *$\mathcal{L}_{SCE}(\boldsymbol{v}^{\star}(\hat{\boldsymbol{y}}), \boldsymbol{y}) \geq 0$*

2. *When the set of optimal solutions is a singleton*
   *$\mathcal{L}_{SCE}(\boldsymbol{v}^{\star}(\hat{\boldsymbol{y}}), \boldsymbol{y}) = 0 \implies Regret(\boldsymbol{v}^{\star}(\hat{\boldsymbol{y}}), \boldsymbol{y}) = 0;$*
   *$Regret(\boldsymbol{v}^{\star}(\hat{\boldsymbol{y}}), \boldsymbol{y}) = 0 \implies \mathcal{L}_{SCE}(\boldsymbol{v}^{\star}(\hat{\boldsymbol{y}}), \boldsymbol{y}) = 0.$*

We also show generalization bounds for SCE loss in Appendix J.

When $\mathcal{L}_{SCE}$ is minimized using a blackbox optimization solver, the gradient would be:

$$\nabla_{\mathcal{L}_{SCE}} = \boldsymbol{v}^{\star} - \boldsymbol{v}^{\star}(\hat{\boldsymbol{y}}) \tag{8}$$

### 3.3 ADDRESSING THE SCALABILITY OF DFL

Implementing DFL entails a significant computational burden, as it requires solving and differentiating the COP or the 'smoothed' COP for each training instance in every epoch using predicted parameters. Mulamba et al. (2021) address the scalability issue by using solution caching instead of repeatedly solving the optimization problem. Tang & Khalil (2024) avoid solving the COP during training using an approach, called CaVE, by minimizing the angle between the predicted cost vector and the 'normal cone' of the true optimal solution, While such approaches aim to avoid solving the COP to improve scalability, a faster and more scalable implementation of the optimization problem is another research direction, which has been receiving increasing attention recently. Research in this area is tangential to the learning-to-optimize paradigm (Bengio et al., 2021; Kotary et al., 2021), which trains an ML model to output COP solutions directly from the parameters.

In order to find a heuristic solution to a COP using ML, graph neural networks emerge as a key building block (Khalil et al., 2022; Cappart et al., 2023). However, for quadratically regularized LPs, a recent work by McKenzie et al. (2024) demonstrate that a simple feedforward neural network can be used to obtain a heuristic solution. Their technique, called DYS-Net, is based on a three-operator splitting technique (Davis & Yin, 2017) to compute the solution. Next, we will provide a brief overview of DYS-Net, as we aim to train by minimizing a surrogate loss using DYS-Net for accelerating DFL.

**DYS-Net for LPs.** The principle of DYS-Net emerges from projected gradient descent (Duchi et al., 2008). Projected gradient descent differs from standard gradient descent in that after each iteration, the current predictions are projected into the feasible space if they are not in the feasible space. However, projecting into the feasible space of a combinatorial optimization problem is itself an expensive operation. If we consider standard form LPs, the feasible space can be expressed as:

$$\mathcal{F} \equiv \mathcal{F}_1 \cap \mathcal{F}_2 \text{ where } \mathcal{F}_1 \doteq \{A\boldsymbol{v} = b\} \text{ and } \mathcal{F}_2 \doteq \{\boldsymbol{v} \geq 0\}.$$

Although projecting an infeasible solution $\boldsymbol{v}$ directly into $\mathcal{F}$ is not a trivial operation, projecting into $\mathcal{F}_1$ and $\mathcal{F}_2$ separately are much simpler tasks. Projecting into $\mathcal{F}_1$ takes the following form:

$$P_{\mathcal{F}_1}(\boldsymbol{v}) \doteq \boldsymbol{v} - A^{\dagger}(A\boldsymbol{v} - b)$$

where $A^{\dagger}$ is the pseudo inverse of $A$. Projecting into $\mathcal{F}_2$ takes the following form:

$$P_{\mathcal{F}_2}(\boldsymbol{v}) \doteq \max\{0, \boldsymbol{v}\}$$

where $\max$ operates element-wise. Cristian et al. (2023) propose an iterative approach, continuously alternating between $P_{\mathcal{F}_1}$ and $P_{\mathcal{F}_2}$. In contrast, DYS-Net introduces the following fixed point iteration algorithm based on a three-operator splitting method.

$$\boldsymbol{v}_{k+1} = \boldsymbol{v}_k - P_{\mathcal{F}_2}(\boldsymbol{v}_k) + P_{\mathcal{F}_1}\Big((2 - \alpha\mu)P_{\mathcal{F}_2}(\boldsymbol{v}_k) - \boldsymbol{v}_k - \alpha\boldsymbol{y}\Big) \tag{DYS}$$

which converges to $\boldsymbol{v}^{\star}(\boldsymbol{y})$ as $k \to \infty$. In practice, we use a finite number of iterations $k$ in a single forward pass to get an approximation of $\boldsymbol{v}^{\star}(\boldsymbol{y})$. Note that all operations in Eq. DYS can be expressed as matrix operations and can be implemented using neural networks, which has the potential for greater scalability and reduced training time by leveraging recent advancements in GPU hardware.

We denote the solution obtained in this method as $DYS(\boldsymbol{y})$. In practice, Eq. DYS would be iterated for a finite number of steps and the resulting solution will be a close approximation to the true optimal outcome. One can view DYS-Net as differentiable optimization by 'neural smoothing'. In case of ILPs, the LP after relaxation is considered. McKenzie et al. (2024) accelerate DF using DYS-Net by minimizing $SqDE$. In this work, we will propose training by minimizing $\mathcal{L}_{SCE}$ or $\mathcal{L}_{SPO+}$ between after replacing $\boldsymbol{v}^\star(\hat{\boldsymbol{y}})$ with $DYS(\hat{\boldsymbol{y}})$ as shown in the next section.

# 4    MINIMIZING SURROGATE LOSS WITH A SMOOTHED SOLVER

Since smoothing converts the non-smooth COP into a smooth differentiable optimization problem, one could compute and differentiate regret using the smoothed problem. Existing works in DFL minimize the empirical regret of the smoothed problem expecting this would reduce the expected regret in unseen instances, aligning with the empirical risk minimization paradigm in ML. However, a close inspection of how the incorporation of smoothing changes the gradient landscape reveals a shortcoming in this approach.

The introduction of smoothing ensures that the solution transitions smoothly, rather than abruptly, near the original COP's transition points. However, the solution of the smoothed optimization remains unchanged, or changes very slowly, in regions where the COP's solution is constant, provided that the smoothing strength is kept low as illustrated in Figure 1b. So in this region, $\frac{d\boldsymbol{v}^\star(\hat{\boldsymbol{y}})}{d\hat{\boldsymbol{y}}}$ is nearly zero. When regret is minimized, the derivative of it with respect to the $\hat{\boldsymbol{y}}$ takes the following form:

$$\left.\frac{\partial \boldsymbol{v}^\star(\boldsymbol{y})}{\partial \boldsymbol{y}}\right|_{\boldsymbol{y}=\hat{\boldsymbol{y}}} \boldsymbol{y} \tag{9}$$

where $\left.\frac{\partial \boldsymbol{v}^\star(\boldsymbol{y})}{\partial \boldsymbol{y}}\right|_{\boldsymbol{y}=\hat{\boldsymbol{y}}}$ is computed by considering the smoothed optimization problem. As we illustrated above, smoothing addresses the non-differentiability at the transition points, but the derivative $\frac{d\boldsymbol{v}^\star(\hat{\boldsymbol{y}})}{d\hat{\boldsymbol{y}}}$ still remains zero far from these points. Hence, the derivative in Eq. 9 remains zero. This would also be true if $SqDE$ is considered as the training loss. In this case, the derivative would be:

$$\left.\frac{\partial \boldsymbol{v}^\star(\boldsymbol{y})}{\partial \boldsymbol{y}}\right|_{\boldsymbol{y}=\hat{\boldsymbol{y}}} (\boldsymbol{v}^\star(\hat{\boldsymbol{y}}) - \boldsymbol{v}^\star) \tag{10}$$

In both Eq. 10 and Eq. 9, the derivative turns zero due to $\left.\frac{\partial \boldsymbol{v}^\star(\boldsymbol{y})}{\partial \boldsymbol{y}}\right|_{\boldsymbol{y}=\hat{\boldsymbol{y}}}$ becoming zero. Consequently, training by gradient descent would fail to change $\hat{\boldsymbol{y}}$ despite $\hat{\boldsymbol{y}}$ resulting non-zero regret.

To prevent the derivative from vanishing far from the transition points, in this paper, we argue in favour of minimizing a surrogate loss instead. For instance, when $\mathcal{L}_{SPO+}$ is minimized, the derivative of the loss with respect to the $\hat{\boldsymbol{y}}$ takes the following form:

$$2(\boldsymbol{v}^\star - \boldsymbol{v}^\star(2\hat{\boldsymbol{y}} - \boldsymbol{y})) + 2\left.\frac{\partial \boldsymbol{v}^\star(\boldsymbol{y})}{\partial \boldsymbol{y}}\right|_{\boldsymbol{y}=2\hat{\boldsymbol{y}}-\boldsymbol{y}} (\boldsymbol{y} - 2\hat{\boldsymbol{y}}) \tag{11}$$

Similarly if $\mathcal{L}_{SCE}$ is minimized after smoothing, the resulting derivative would be:

$$(\boldsymbol{v}^\star - \boldsymbol{v}^\star(\hat{\boldsymbol{y}})) + \left.\frac{\partial \boldsymbol{v}^\star(\boldsymbol{y})}{\partial \boldsymbol{y}}\right|_{\boldsymbol{y}=\hat{\boldsymbol{y}}} (\boldsymbol{y} - \hat{\boldsymbol{y}}) \tag{12}$$

The way Eq. 12 differs from Eq. 9 is the term $(\boldsymbol{v}^\star - \boldsymbol{v}^\star(\hat{\boldsymbol{y}}))$ and the multiplier of $\left.\frac{\partial \boldsymbol{v}^\star(\boldsymbol{y})}{\partial \boldsymbol{y}}\right|_{\boldsymbol{y}=\hat{\boldsymbol{y}}}$ is $(\boldsymbol{y} - \hat{\boldsymbol{y}})$ instead of $\boldsymbol{y}$. The term $(\boldsymbol{v}^\star - \boldsymbol{v}^\star(\hat{\boldsymbol{y}}))$ prevents $\frac{d\mathcal{L}}{d\hat{\boldsymbol{y}}}$ going to zero even when $\frac{d\boldsymbol{v}^\star(\hat{\boldsymbol{y}})}{d\hat{\boldsymbol{y}}} \approx 0$. Note that if we minimize $\mathcal{L}_{SCE}$ or $\mathcal{L}_{SPO+}$ using a blackbox optimization solver, $\frac{\partial \boldsymbol{v}^\star(\boldsymbol{y})}{\partial \boldsymbol{y}}$ cannot be computed and only the first part of the derivative would be used. So, in this case, Eq. 11 and Eq. 12 would reduce to Eq. 5 and Eq. 8 respectively.

In principle, both $\mathcal{L}_{SCE}$ or $\mathcal{L}_{SPO+}$ can be minimized using a 'smoothed' solver. However, there is an important distinction to note. $\mathcal{L}_{SCE}$ is non-convex in $\hat{\boldsymbol{y}}$ whereas $\mathcal{L}_{SPO+}$ is convex. On the other hand, $\mathcal{L}_{SPO+}$ can be non-zero even when regret is zero (see Appendix C), whereas Proposition 1 shows that $\mathcal{L}_{SCE}$ is zero whenever regret is zero. As smoothing can potentially improve its performance by

eliminating the sharp transitions in $\mathcal{L}_{SCE}$ as per Proposition 1, and $\mathcal{L}_{SCE}$ is closely related to regret, minimizing $\mathcal{L}_{SCE}$ after smoothing may be a promising approach, provided that smoothing does not significantly alter the solution. We will empirically investigate whether $\mathcal{L}_{SCE}$ or $\mathcal{L}_{SPO+}$ fits well with a smoothed solver.

**A deep dive into the gradient landscape.** To convince readers that the solution of the smoothed optimization remains unchanged, we will demonstrate how the gradient landscape changes after QP smoothing with a simple illustration. For this, we consider the following one-dimensional optimization problem:

$$\min_{v} yv \text{ s.t. } 0 \le v \le 1 \tag{13}$$

where $y \in \mathbb{R}$ is the parameter to be predicted. Note that the solution of this problem is: $v^{\star}(y) = 1$ if $y < 0$ and $v^{\star}(y) = 0$ if $y > 0$. When $y = 0$ any value in the interval [0,1] is an optimal solution.

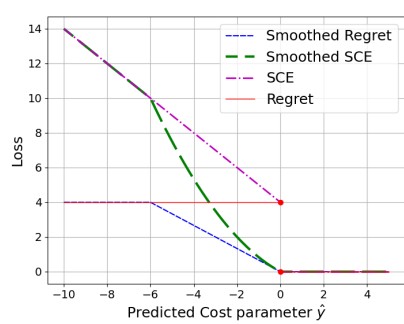

Figure 2: The numerical illustration demonstrates that while smoothing removes abrupt changes in the solution and makes the regret continuous, the solution often remains flat across most regions, resulting in a zero gradient, not suitable for training. In contrast, $\mathcal{L}_{SCE}$ (with or without smoothing) provides a more responsive landscape: when regret is non-zero, $\mathcal{L}_{SCE}$ ensures non-zero gradient.

Let us assume that the true value of $y$ is $4$ and hence $v^{\star}(y) = 0$. The red line in Figure 2 shows how the value of regret changes as $\hat{y}$ changes. The regret is $4$ when $\hat{y} \le 0$ and $0$ when $\hat{y} > 0$. The regret changes abruptly at the point $\hat{y} = 0$. After augmenting the objective with the quadratic smoothing term $\frac{\mu}{2}v^2$ with $\mu > 0$, the solution of the smoothed problem is:

$$v^{\star}(y) = \begin{cases} 0; & \text{when } y > 0 \\ -\frac{y}{\mu}; & \text{when } -\mu < y \le 0 \\ 1; & \text{when } y \le -\mu \end{cases}$$

This makes the derivative non-zero in the interval $-\mu \le y \le 0$. However, it is still zero when $y < -\mu$. Hence, if $\hat{y} < -\mu$, *the derivative of regret is $0$, even if regret is non-zero.* Consequently, the predictions cannot be changed by gradient descent despite regret being zero. The regret with the smoothed problem is shown by the blue line in Figure 2 for $\mu = 6$. The strength of smoothing can be increased by assigning $\mu$ to a high value. However, if $\mu \gg |y|$, $v^{\star}(y) \approx 0$ almost everywhere. On the other hand, $\mathcal{L}_{SCE}$ with and without smoothing are plotted with green and violet colors, respectively. In both cases, $\mathcal{L}_{SCE}$ is strictly decreasing for $\hat{y} < 0$. This ensures a non-zero derivative, suitable for guiding $\hat{y}$ towards the positive

half-space if $\hat{y} < 0$. Minimizing regret generates a zero gradient, preventing effective gradient-based learning, as explained with an example in Appendix D. Moreover, we have performed computational experiments in Appendix B on larger optimization problems to show that the zero-gradient problem persists in optimization problems with a large number of parameters and decision variables.

## 5 EXPERIMENTAL EVALUATION

In order to demonstrate the advantage of using $\mathcal{L}_{SCE}$ as the training loss, we report experiments on three well-established DFL benchmark problems.

**Multi-Dimensional Knapsack (KP).** The objective of the knapsack problem is to select a subset of items with the highest total value, subject to a capacity constraint. The weights of the items and the knapsack's capacity are known, but the values of the items are unknown. Therefore, the prediction task is to predict the value of each item using features.

**Shortest path on a grid (SP).** The goal of this optimization problem is to find the path, with lowest cost on a $k \times k$ grid, starting from the southwest node and ending at the northeast node of the grid (Elmachtoub & Grigas, 2022). The cost of each edge is unknown and should be predicted before

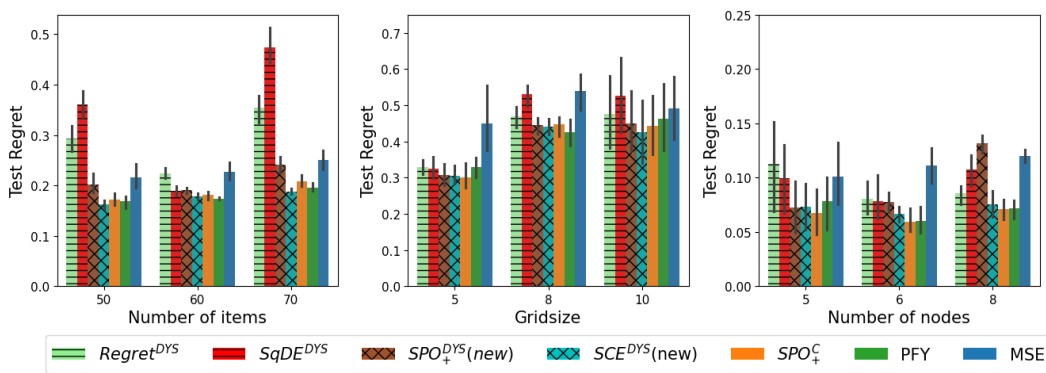

Figure 3: Experiment with DYS-Net in relatively smaller KP, SP and TSP instances (from left to right). $SPO_+^C$ minimizes $\mathcal{L}_{SPO+}$ using Gurobi solvers. When 'DYS' appears as a superscript of a loss, it indicates that the loss is computed and minimized using DYS-Net.

solving the problem. The true relation between the features and the costs are non-linear, but linear model is used for predictions.

**Travelling salesperson problem (TSP).** Given a set of nodes, the goal is to find the tour, with the lowest cost, that visits every node exactly once. As before, the costs are related to the features in a non-linear manner, but a linear predictive model is used for prediction.

We use *PyEPO* (Tang & Khalil, 2023) to generate the training, validation and test instances for the SP, KP and TSP problems. In all three problems, **the true relation between the features and the costs are non-linear, but linear model is used for predictions.** We experiment with polynomial degree parameter and noise half-width parameter being 6 and 0.5, respectively. The predictive models are implemented using *PyTorch* (Paszke et al., 2019) and *Gurobipy* (Gurobi Optimization, 2021) is used as a combinatorial solver to obtain the optimal solution. For evaluation, we always use *Gurobipy*. For all the experiments, We repeat each experiment five times and report **normalized relative regret** on test data, calculated as follows:

$$\frac{1}{N_{test}} \sum_{i=1}^{N_{test}} \frac{\boldsymbol{y}_i^\top (\boldsymbol{v}^\star(\hat{\boldsymbol{y}}_i) - \boldsymbol{v}_i^\star)}{\boldsymbol{y}_i^\top \boldsymbol{v}_i^\star}. \tag{14}$$

We use the implementation of DYS-NET by McKenzie et al. (2024). The experiments were executed on a computer with an *Intel(R) Core(TM) i7-13800H* processor using 32 Gb of RAM.

## 5.1 EXPERIMENT WITH DYS-NET

**RQ1: Are surrogate losses better suited for DYS-Net?** In the first set of experiments, we want to investigate whether minimizing the surrogate losses result in regret lower than minimizing $Regret$ and $SqDE$. For this set of experiments, we consider relatively small-sized COPs. We use three approaches as benchmarks: A prediction-focused approach, $MSE$, which minimizes the MSE loss between $\boldsymbol{y}$ and $\hat{\boldsymbol{y}}$; a DFL implementation of perturbed Fenchel-Young (PFY) (Berthet et al., 2020) loss; and another DFL approach, which minimizes $\mathcal{L}_{SPO+}$ by solving the COPs. We show the normalized regret across five runs in Figure 3. It is clearly evident across all problems that minimizing $\mathcal{L}_{SCE}$ and $\mathcal{L}_{SPO+}$ results in lower regret than minimizing $Regret$ and $SqDE$. This set of experiments highlight the advantage of minimizing surrogate losses with DYS-Net.

**RQ2: Does DYS-Net accelerates DFL?** We present the result for larger problem instances of KP, SP and TSP in Figure 4. In the upper and lower panels, we compare normalized relative test regret and training time of one epoch, respectively. For these experiments, we did not report regret of PFY, because its performance is same as $SPO_+^C$. Moreover, we did not consider minimizing $Regret$ and $SqDE$ with DYS-Net, because previous experiments reveal they produce higher regret. As, training time is the primary focus, we include the DFL approaches, which are focused on that: CaVE and minimizing $\mathcal{L}_{SPO+}$ with solution-caching.

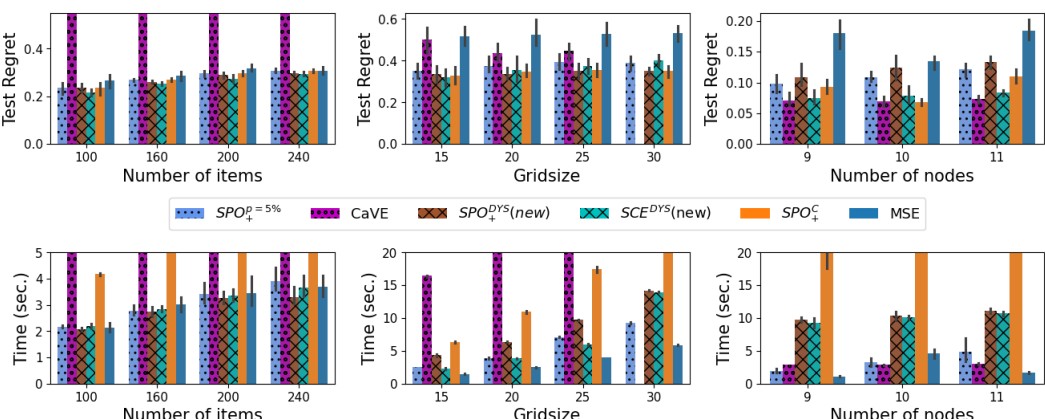

Figure 4: Experiment with DYS-Net in relatively larger KP, SP and TSP instances (from left to right). $SPO_+^{p=5\%}$ minimizes $\mathcal{L}_{SPO+}$ using using a solution cache, with $p_{solve} = 5\%$. $CaVE$ minimizes the the negative of cosine similarity between $\hat{y}$ and the optimal cone.

In terms of test regret, in all the KP and TSP instances, $\mathcal{L}_{SCE}$ with DYS-Net, $SCE^{DYS}$, matches the regret of the $SPO_+^C$ approach with significant reduction in training time. For the shortest path instances up to grid-size of 25, $SCE^{DYS}$, produces regret comparable to $SPO_+^C$; however, regret increases for grid-size of 30.

In summary, $SCE^{DYS}$ yields regret similar to $SPO_+^C$, while significantly reducing runtime. The advantage becomes more pronounced with larger problem sizes; for instance, in the 11-node TSP, DYS-Net is 5 times faster than SPO, which solves an ILP. Notably, these results were achieved without GPU training, suggesting that even greater runtime reductions are possible with GPU usage.

The solution-caching approach, $SPO_+^{p=5\%}$, is faster than DYS-Net especially in the TSP instances. With respect to test regret, $SPO_+^{p=5\%}$ performs poorer than $SCE^{DYS}$ in the TSP instances, however for other instances its regret comparable to $SCE^{DYS}$. We have also tested against $SCE^{p=5\%}$, which performs slightly worse (see Figure 9 in Appendix F.)

**Comparison against CaVE.** CaVE performs well for TSPs in terms of regret and time but struggles with other problems. For TSPs, the relatively small number of active constraints makes the method efficient. However, for SP problems, the exponential growth in active constraints causes memory overhead issues, as all active constraints are stored in memory. For instance, a 10-node TSP has 100 active constraints, while a 10-node SP has 200. Due to memory overhead, we cannot run CaVE on the 30-grid SP instance. Thus, we can say CaVE is suitable for TSPs, but not generalizable to all optimization problems. Moreover, another shortcoming of CaVE is visible in the KP problem, where it fails in terms of both quality and scalability. This is because their method relies on identifying active constraints, but the capacity constraint in the integer knapsack problem is often not active. This leads to incorrect identification of the normal cone resulting in poor performance of CaVE in the knapsack problem (see Appendix M for further explanation).

**RQ3: Evaluation on larger TSP instances.** In the next set of experiments, we aim to compare $SCE^{DYS}$ against CaVE and $SPO_+^{p=5\%}$ on even larger TSP instances. One advantage of CaVE is that it uses the DFJ formulation, whereas DYS-Net relies on the MTZ formulation, as the latter requires specifying the full problem. Since the MTZ formulation is weaker than the DFJ formulation (Öncan et al., 2009; Langevin et al., 1990), we adapted DYS-Net by drawing inspiration from CaVE. Specifically, like CaVE, we first identify and collect the active constraints for all instances before starting DFL training. Instead of considering the true ILP representing the original TSP, we consider an ILP constructed only from these active constraints. Then, rather than solving the quadratically relaxed LP of the original TSP, we solve the relaxed LP derived from the ILP of the active constraints using DYS-Net. It is important to note that this adaptation results in a different ILP for each instance,

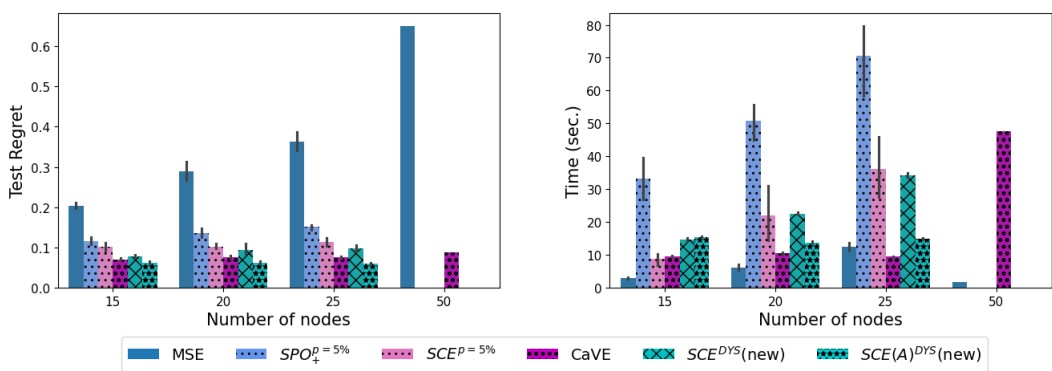

Figure 5: Experiment with DYS-Net in larger TSP instances.

as the active constraints corresponding to the true solution vary across instances. This adaptation of is denoted by $SCE(A)^{DYS}$ (where $A$ indicates active constraints).

In Figure 5, we consider TSP instances with 15, 20 and 25 nodes. For these larger problem instances, we cannot complete training of $SPO_+^C$. We focus exclusively on TSP instances because, among the three optimization problems considered it is the most difficult and time-consuming to solve. First, we point out that the training time of $SCE(A)^{DYS}$ is significantly lower than $SCE^{DYS}$. because it solves a smaller problem. Secondly, as problem size increases, DYS-Net demonstrates better scalability than $SCE^{p=5\%}$ and $SPO_+^{p=5\%}$ since solving the optimization problem even for 5% of the training instances becomes significantly time-intensive. The reason for the discrepancy between $SCE^{p=5\%}$ and $SPO_+^{p=5\%}$ is explained in Appendix G. CaVE, on the other hand, is proved to be faster than both $SCE^{DYS}$ and $SCE(A)^{DYS}$, although $SCE(A)^{DYS}$ reduces the gap quite significantly. Moreover, $SCE(A)^{DYS}$ yields lower regret than CaVE.

**Summary.** We first demonstrated that minimizing $\mathcal{L}_{SCE}$ and $\mathcal{L}_{SPO+}$ achieves lower regret than directly minimizing $Regret$ and $SqDE$ using DYS-Net. Notably, Table 3 in the Appendix O confirms that this result also holds for other smoothed differentiable solvers, such as *Cvxpylayer*. Next, we have shown that the test regret of $SCE^{DYS}$ is comparable to PFY and $SPO_+^C$, while $SCE^{DYS}$ requires significantly reduced training time. In terms of training time, CaVE is faster than $SCE^{DYS}$ for the TSP problems. Moreover, inspired by CaVE, we adapted $SCE^{DYS}$ requiring solving a smaller ILP. Although this new approach is not faster than CaVE, it results in lower regret. Lastly, while CaVE has a lower training time than $SCE(A)^{DYS}$ and $SCE^{DYS}$, we found that it performs poorly on other problems. In contrast, $SCE^{DYS}$ is a scalable DFL approach and it is applicable to a broad class of ILPs.

## 6 CONCLUSION

In this paper, we experiment with the recently proposed DYS-Net, a fast neural solver for LPs. By minimizing $Regret$ or $SqDE$, DYS-Net cannot attain regret as low as existing DFL techniques, such as SPO and PFY. So, we challenge the conventional DFL approach of directly minimizing empirical regret when a smoothing operation is applied to make the optimization problem differentiable. Instead, we recommend minimizing a surrogate loss, such as $\mathcal{L}_{SCE}$ and justify this by comparing the pattern of the gradient landscape concerning regret and the surrogate loss. By doing so, we effectively merge the two families of approaches in DFL. Our experimental evaluations show that for most problems minimizing $\mathcal{L}_{SCE}$ using DYS-Net produces regret as low as the state-of-the-art SPO method, with a clear advantage in runtime up to five-fold.

Future work includes applying this approach to real-world large-scale applications with full GPU training. Furthermore, new fully neural smoothing approaches or better surrogate losses can also benefit from this joint approach. While used here for linear objective functions, future work can investigate the joint applicability of both smoothing and surrogates for non-linear optimization too.

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

## A    Proof of Proposition 1

*Proof.*    1. Following the definition of $\mathcal{L}_{SCE}$,

$$\mathcal{L}_{SCE}(\boldsymbol{v}^\star(\hat{\boldsymbol{y}}), \boldsymbol{y}) = (\hat{\boldsymbol{y}} - \boldsymbol{y})^\top \boldsymbol{v}^\star(\boldsymbol{y}) - (\hat{\boldsymbol{y}} - \boldsymbol{y})^\top \boldsymbol{v}^\star(\hat{\boldsymbol{y}})$$
$$= \hat{\boldsymbol{y}}^\top (\boldsymbol{v}^\star(\boldsymbol{y}) - \boldsymbol{v}^\star(\hat{\boldsymbol{y}})) + \boldsymbol{y}^\top (\boldsymbol{v}^\star(\hat{\boldsymbol{y}}) - \boldsymbol{v}^\star(\boldsymbol{y}))$$

$\hat{\boldsymbol{y}}^\top (\boldsymbol{v}^\star(\boldsymbol{y}) - \boldsymbol{v}^\star(\hat{\boldsymbol{y}})) \geq 0$, because $\boldsymbol{v}^\star(\hat{\boldsymbol{y}})$ is the optimal solution to $\hat{\boldsymbol{y}}$. In a similar way, $\boldsymbol{y}^\top (\boldsymbol{v}^\star(\hat{\boldsymbol{y}}) - \boldsymbol{v}^\star(\boldsymbol{y})) \geq 0$. Hence, $\mathcal{L}_{SCE}(\boldsymbol{v}^\star(\hat{\boldsymbol{y}}), \boldsymbol{y}) \geq 0$.

2. We will prove the claim by contradiction. Assume that $\mathcal{L}_{SCE}(\boldsymbol{v}^\star(\hat{\boldsymbol{y}}), \boldsymbol{y}) = 0$ but $Regret(\boldsymbol{v}^\star(\hat{\boldsymbol{y}}), \boldsymbol{y}) = \boldsymbol{y}^\top (\boldsymbol{v}^\star(\hat{\boldsymbol{y}}) - \boldsymbol{v}^\star(\boldsymbol{y})) > 0$ . This is possible if $\boldsymbol{v}^\star(\hat{\boldsymbol{y}}) \neq \boldsymbol{v}^\star(\boldsymbol{y})$. As the solution to $\hat{\boldsymbol{y}}$ is different from $\boldsymbol{v}^\star(\boldsymbol{y})$, the singleton assumption implies that $\exists\, \boldsymbol{v}' \in \mathcal{F} \setminus \{\boldsymbol{v}^\star(\boldsymbol{y})\} : \hat{\boldsymbol{y}}^\top \boldsymbol{v}' < \hat{\boldsymbol{y}}^\top \boldsymbol{v}^\star(\boldsymbol{y})$. In this case, we have:

$$\hat{\boldsymbol{y}}^\top \boldsymbol{v}^\star(\boldsymbol{y}) - \hat{\boldsymbol{y}}^\top \boldsymbol{v}' > 0$$
$$\Rightarrow (\hat{\boldsymbol{y}}^\top \boldsymbol{v}^\star(\boldsymbol{y}) - \hat{\boldsymbol{y}}^\top \boldsymbol{v}') + (\boldsymbol{y}^\top \boldsymbol{v}' - \boldsymbol{y}^\top \boldsymbol{v}^\star(\boldsymbol{y})) > (\boldsymbol{y}^\top \boldsymbol{v}' - \boldsymbol{y}^\top \boldsymbol{v}^\star(\boldsymbol{y})) \geq 0$$
$$\Rightarrow (\hat{\boldsymbol{y}} - \boldsymbol{y})^\top \boldsymbol{v}^\star - (\hat{\boldsymbol{y}} - \boldsymbol{y})^\top \boldsymbol{v}^\star(\hat{\boldsymbol{y}}) > 0$$

In the second line, $\boldsymbol{y}^\top \boldsymbol{v}' - \boldsymbol{y}^\top \boldsymbol{v}^\star(\boldsymbol{y})$ is added in both sides and this term is nonnegative as $\boldsymbol{v}^\star(\boldsymbol{y})$ is the optimal solution to $\boldsymbol{y}$. This implies $\mathcal{L}_{SCE}(\boldsymbol{v}^\star(\hat{\boldsymbol{y}}), \boldsymbol{y}) > 0$ and we arrive at a contradiction. Thus we prove that $\mathcal{L}_{SCE}(\boldsymbol{v}^\star(\hat{\boldsymbol{y}}), \boldsymbol{y}) = 0 \implies Regret(\boldsymbol{v}^\star(\hat{\boldsymbol{y}}), \boldsymbol{y}) = 0$.

Next, assume $Regret(\boldsymbol{v}^\star(\hat{\boldsymbol{y}}), \boldsymbol{y}) = 0$. This implies that $\boldsymbol{y}^\top \boldsymbol{v}^\star(\hat{\boldsymbol{y}}) = \boldsymbol{y}^\top \boldsymbol{v}^\star(\boldsymbol{y})$. This can only be true if $\boldsymbol{v}^\star(\hat{\boldsymbol{y}}) = \boldsymbol{v}^\star(\boldsymbol{y})$ because of the singleton assumption. Hence, $\mathcal{L}_{SCE}(\boldsymbol{v}^\star(\hat{\boldsymbol{y}}), \boldsymbol{y}) = (\hat{\boldsymbol{y}} - \boldsymbol{y})^\top (\boldsymbol{v}^\star(\boldsymbol{y}) - \boldsymbol{v}^\star(\hat{\boldsymbol{y}})) = 0$.

$\square$

## B    Computational Experiments Demonstrating Zero Gradient

In Section 4, we made the case for minimizing surrogate loss such as $\mathcal{L}_{SCE}$ instead of $Regret$. Our main argument is for a relatively low value of smoothing parameter $\mu$, $Regret$ will have zero gradient. However, $\mathcal{L}_{SCE}$ will not have this problem. We provided two illustrations considering small-scale optimization problems. In this case, we justify this with higher-dimensional optimization problems. We consider Top-1 selection problem with different number of items $M$.

$$\max_{\boldsymbol{v} \in \{0,1\}} \boldsymbol{y}^\top \boldsymbol{v} \text{ s.t. } \boldsymbol{v}^\top \mathbf{1} \leq 1 \tag{15}$$

| | | | M | | | |
|---|---|---|---|---|---|---|
| $\mu$ | 5 | 10 | 20 | 40 | 80 | 100 |
| 0.100 | 0.000 | 0.000 | 0.000 | 0.000 | 0.000 | 0.000 |
| 0.500 | 0.000 | 0.000 | 0.000 | 0.000 | 0.000 | 0.000 |
| 0.990 | 0.000 | 0.000 | 0.000 | 0.000 | 0.000 | 0.001 |
| 1.050 | 0.089 | 0.089 | 0.089 | 0.089 | 0.089 | 0.089 |
| 1.500 | 0.465 | 0.466 | 0.465 | 0.465 | 0.465 | 0.464 |
| 2.000 | 0.622 | 0.622 | 0.622 | 0.622 | 0.622 | 0.622 |
| 5.000 | 1.165 | 1.165 | 1.165 | 1.165 | 1.165 | 1.165 |

Table 1: We tabulate average Manhattan distance between the solution of the 'smoothed' problem and the solution of the original problem for different values of $M$ and $\mu$.

$\boldsymbol{y} = [y_1, \ldots, y_M] \in \mathbb{R}^M$ is the vector denoting value of all the items and $\boldsymbol{v} = [v_1, \ldots, v_M]$ is the vector decision variables. To replicate the setup of a PtO problem, we solve the optimization problem with $\hat{\boldsymbol{y}}$. Let us assume $y_i, \hat{y}_i \geq 0$.

Before, solving the problem with simulation, we will show one interesting aspect of this problem. Note that when $\mu > 0$, the following relaxed optimization problem is solved:

$$\max_{\boldsymbol{v}} \boldsymbol{y}^\top \boldsymbol{v} - \frac{\mu}{2}||\boldsymbol{v}||^2 \text{ s.t. } \boldsymbol{v}^\top \mathbf{1} \leq 1; \;\; \boldsymbol{v} \geq 0 \tag{16}$$

We point out that the solution to the unconstrained optimization problem is $v_i^\star = \frac{y_i}{\mu} > 0$.

The augmented Lagrangian of Equation 16 is

$$\mathbb{L} = \boldsymbol{y}^\top \boldsymbol{v} - \frac{\mu}{2}||\boldsymbol{v}||^2 + \lambda(1 - \boldsymbol{v}^\top \mathbf{1}) + \boldsymbol{\sigma}^\top \boldsymbol{v} \tag{17}$$

where $\lambda$ and $\boldsymbol{\sigma} = [\sigma_1, \ldots, \sigma_M]$ are dual variables. By differentiating $\mathbb{L}$ with respect to $v_i$, we obtain one condition of optimality, which is the following:

$$y_i - \mu v_i - \lambda + \sigma_i = 0 \implies v_i = \frac{y_i - \lambda + \sigma_i}{\mu} \tag{18}$$

Without any loss of generality, let $y^{(1)} \geq y^{(2)} \geq \ldots y^{(M)}$. (In the d) As, solution to the constrained optimization problem is $v_i > 0$, $y^{(1)}$ will definitely be greater than zero. Hence, $\sigma_i = 0$ because of strict complementarity. So, we can write $v^{(1)} - v^{(k)} = \frac{y^{(1)} - y^{(k)} - \sigma^{(k)}}{\mu}$. As, $v^{(1)} - v^{(k)} \leq 1$, we can write:

$$\frac{y^{(1)} - y^{(k)} - \sigma^{(k)}}{\mu} \leq 1 \implies \mu \geq y^{(1)} - y^{(k)} - \sigma^{(k)} \tag{19}$$

So,

$$y^{(1)} - y^{(k)} > \mu \implies \sigma^{(k)} > 0 \implies y^{(k)} = 0 \tag{20}$$

This suggest that if $y^{(k)} < y^{(1)} - \mu$, only $v^{(1)} = 1$ and all other decision variables will be zero in the optimal solution.

To generate the ground truth $\boldsymbol{y}$, we randomly select $M$ integers without replacement from the set $1, \ldots, M$. The predicted costs, $\hat{\boldsymbol{y}}$, are generated by considering a different sample from the same set. As a result, $\boldsymbol{y}$ and $\hat{\boldsymbol{y}}$ contain the same numbers but in different permutations. It is important to note that all elements in both vectors are positive integer values. We compute the solution to the optimization problem for $\boldsymbol{y}$ and $\hat{\boldsymbol{y}}$. We solve the optimization problem with $\hat{\boldsymbol{y}}$ using a 'smoothed' optimization layer—*CvxpyLayer*. in order to compare the gradients of *Regret* and $\mathcal{L}_{SCE}$. We compute the gradients of both the losses for multiple values of $M$ and $\mu$. For each configuration of $M$ and $\mu$, we run 20 simulations.

Note that $y^{(1)} > y^{(2)} > \ldots y^{(M)}$ because of the way we created the dataset. Moreover, as all values in $\hat{\boldsymbol{y}}$ and $\boldsymbol{y}$ are integer, Equation 20 suggests if $\mu < 1$, the solution to the relaxed problem

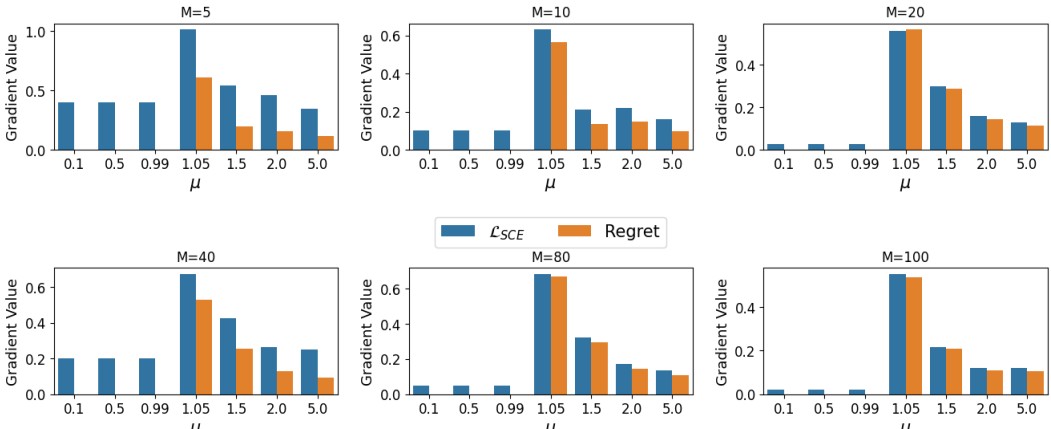

Figure 6: Results of Computational Simulation

(equation 16) will be binary. So, the discussion in Section 4 suggests that slight change of the cost parameter would not change the solution and hence the zero gradient problem would appear while differentiating $Regret$.

In Figure 6, we plotted the average absolute values of the gradients of the two losses— $\mathcal{L}_{SCE}$ and $Regret$. As we hypothesized the gradient turns zero whenever $Regret$ is minimized with $\mu < 1$. It is true that for $\mu > 1$, $Regret$ have non-zero gradient. However, higher values of $\mu$ turns solution to the 'smoothed' problem very different from the solution to the original problem. We show this in Table 1 by displaying the average Manhattan distance between solutions of the true and 'smoothed' problem for same $\hat{y}$.

We also highlight that, for the same values of $\mu$, the average Manhattan distances remain same across different $M$. Examining the results of the simulations, we observed that the solution to the smoothed problem is fractional. For example, when $\mu = 2$, the solution includes two non-zero values— 0.77 and 0.23. Typically, the value 0.77 appears in the position corresponding to the highest value in $\hat{y}$, i.e., where there is a 1 in solution vector. As a result, the Manhattan distance becomes (1-0.77)+0.23 = 0.46. Interestingly, these values remain unchanged across different values of $M$. Therefore, the Manhattan distance remains constant as long as $\mu$ does not change.

## C    COMPARISON BETWEEN $\mathcal{L}_{SPO+}$ AND $\mathcal{L}_{SCE}$

The SPO+ loss, $\mathcal{L}_{SPO+}(\boldsymbol{v}^{\star}(\hat{\boldsymbol{y}}), \boldsymbol{y})$, proposed by Elmachtoub & Grigas (2022) is a convex function of $\hat{\boldsymbol{y}}$. However, the $\mathcal{L}_{SCE}$ loss proposed by Mulamba et al. (2021) is non-convex with respect to $\hat{\boldsymbol{y}}$. Note that,

$$\mathcal{L}_{SCE}(\boldsymbol{v}^{\star}(\hat{\boldsymbol{y}}), \boldsymbol{y}) = \hat{\boldsymbol{y}}^{\top}(\boldsymbol{v}^{\star}(\boldsymbol{y}) - \boldsymbol{v}^{\star}(\hat{\boldsymbol{y}})) + \boldsymbol{y}^{\top}(\boldsymbol{v}^{\star}(\hat{\boldsymbol{y}}) - \boldsymbol{v}^{\star}(\boldsymbol{y}))$$

We can easily show the convexity of $\mathcal{L}_{SCE}$ with a numerical example. Let us consider the example introduced in Equation 13. In Figure 7, we plot $\mathcal{L}_{SCE}$ and $\mathcal{L}_{SPO+}$ for different values of $\hat{y}$. To make this plot, we use an exact solver, not the 'smoothed' solver. Note that, $\mathcal{L}_{SCE}$ includes a jump when the solution of $\hat{y}$ switches from 1 to 0. However, this is not the case for $\mathcal{L}_{SPO+}$. More specifically, $\frac{3}{4}\mathcal{L}_{SCE}(2, y) + \frac{1}{4}\mathcal{L}_{SCE}(-2, y) > \mathcal{L}_{SCE}(\frac{3}{4}(2) + \frac{1}{4}(-2), y) = \mathcal{L}_{SCE}(1, y)$, which violates the definition of a convex function.

We also point out that $\mathcal{L}_{SPO+}$ is can be non-zero, even if regret is zero. Note in Figure 7, for $\hat{y} \in (0, 2)$ regret is zero, but $\mathcal{L}_{SPO+}$ is not. However, you can see $\mathcal{L}_{SCE}$ is zero if regret is zero (also proved in Proposition 1).

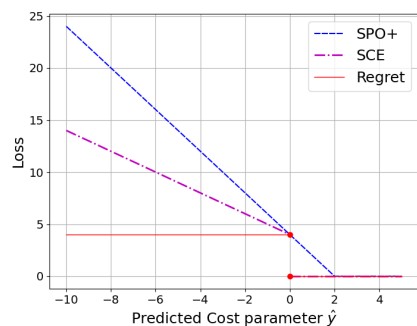

Figure 7: A numerical illustration to show $\mathcal{L}_{SCE}$ is not convex, but $\mathcal{L}_{SPO+}$ is.

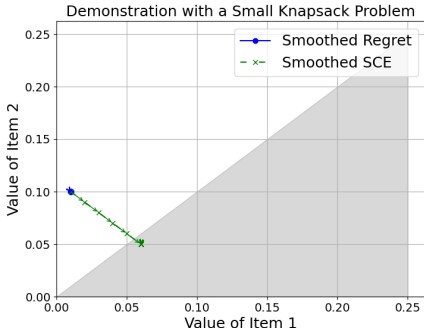

Figure 8: Progression of predictions by epochs when the smoothed regret and SCE are used as training losses.

## D    DEMONSTRATION OF LEARNING WITH $\mathcal{L}_{SCE}$ VERSUS REGRET

We further illustrate this with a simple fractional knapsack problem, which is an LP. Let us consider that we have two items and space for only one item. This can be formulated as a minimization problem:

$$\min \quad -y_1 v_1 - y_2 v_2 \quad \text{s.t.} \quad v_1 + v_2 \leq 1; \quad v_1, v_2 \geq 0$$

Let us assume the true values of $y_1$ and $y_2$ are $(0.8, 0.4)$. The corresponding solution is $(v_1, v_2) = (1, 0)$. The grey region in Figure 8 corresponds to any predictions satisfying $\hat{y}_1 > \hat{y}_2$. Such predictions will induce the true solution, resulting in zero regret. Further assume that the initial predictions are $(\hat{y}_1, \hat{y}_2) = (0.1, 0.01)$. We show the progression of predictions by epochs when regret and SCE are used as training loss, using the smoothed optimization problem with blue and green lines, respectively in Figure 8. The predictions does not change with training epochs when regret is used as the loss because the derivatives of regret with respect to $\hat{y}_1$ and $\hat{y}_2$ are zero. On the other hand, when $\mathcal{L}_{SCE}$ is used as the loss, $(\hat{y}_1, \hat{y}_2)$ gradually move from the white region to the grey region, eventually resulting in zero regret. Note that increasing the strength of smoothing may provide non-zero gradient across the space. But this will entirely alter the optimization problem's solution. For instance, in this knapsack example, high values of $\mu$ would make both $v_1$ and $v_2$ close to zero.

## E    MINIMIZING $\mathcal{L}_{SPO+}$ USING DYS-NET

In Table 3, we show that minimizing $\mathcal{L}_{SCE}$ results in lower regret compared to minimizing $\mathcal{L}_{SPO+}$ using *CvxpyLayer*. Since both *CvxpyLayer* and DYS-Net are differentiable 'smoothed' layers, we would expect similar results with DYS-Net. For this reason, we included only $\mathcal{L}_{SCE}$ in Figure 3. To ensure completeness, we added the results of minimizing $\mathcal{L}_{SPO+}$ with DYS-Net in Figure **??**. As we hypothesized, this leads to higher average regret compared to minimizing $\mathcal{L}_{SCE}$.

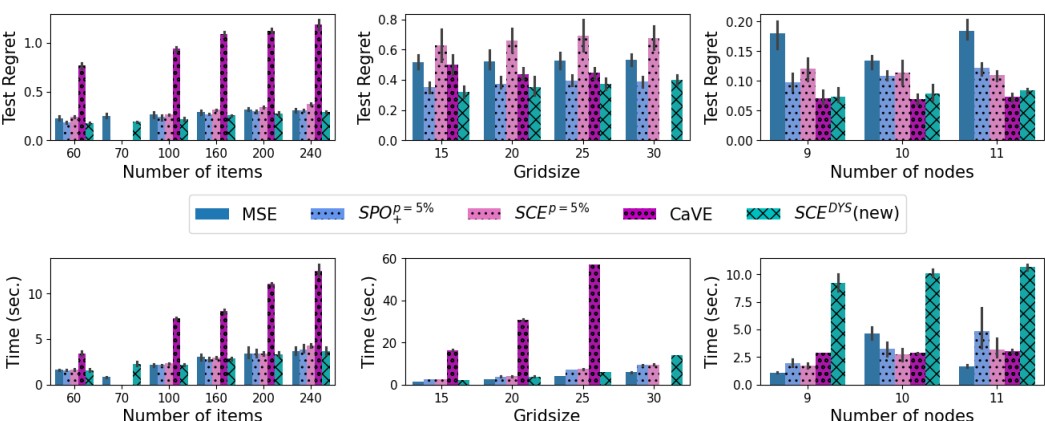

Figure 9: Comparison between DYS-Net and solution caching and CaVE.

## F  COMPARISON AGAINST SOLUTION CACHING

To reduce the long training time of DFL, Mulamba et al. (2021) propose the idea of solution caching. Instead of finding the optimal solution to $\hat{y}$ or $((2\hat{y} - y)$ for $\mathcal{L}_{SPO+})$, Mulamba et al. (2021) suggest returning a heuristic solution by selecting the optimal one from a finite-dimensional 'cache.' They initialize the cache with all existing solutions in the training data. Furthermore, during training, they randomly solve for $p\%$ of the training instances. Note that if, solve ratio, $p = 100\%$, this strategy becomes equivalent to solving the combinatorial problem for every instance. Conversely, if $p = 0\%$, no additional problem-solving is required during training.

We compare the performance of DYS-Net with solution caching in Figure 9. $SPO_+^{p=10\%}$ denotes the case where $\mathcal{L}_{SPO+}$ is minimized with a solve ratio of 10%. Similarly, $SPO_+^{p=5\%}$ and $SPO_+^{p=0\%}$ correspond to solve ratios of 5% and 0%, respectively. Similarly, $SCE^{p=5\%}$ stands for minimizing $\mathcal{L}_{SCE}$ with $p = 5\%$. Note that while solution caching approach, Equation 5 and Equation 8 are used for backpropagating $\mathcal{L}_{SPO+}$ and $\mathcal{L}_{SCE}$ respectively.

It is evident in Figure 9 that $p = 0\%$ results in higher regret for $\mathcal{L}_{SPO+}$. However, the regret is much lower for $p$ being 5% and 10%. Nevertheless, we point out minimizing $\mathcal{L}_{SCE}$ with DYS-Net results in lower regret. This is particularly prominent for the TSP instances. In terms of training efficiency, solution caching has lower training time for these instances.

## G  COMPARATIVE ANALYSIS IN LARGER TSP INSTANCES

In Figure 3, we compared TSP instances till 11 nodes. This is due to the fact that for larger TSP instances, we cannot complete training of $SPO_+^{combinatorial}$ and $SCE^{cvxpy}$. In Figure 10, we consider TSP instances with 15, 20 and 25 nodes. We focus exclusively on TSP instances because, among the three optimization problems considered, because it is the most difficult and time consuming to solve. We have excluded $SPO_+^{combinatorial}$ and $SCE^{cvxpy}$ and included $SPO_+^{p=5\%}$ and $SCE^{p=5\%}$.

We first draw the reader's attention to the observation that $SPO_+^{p=5\%}$ requires more training time compared to $SCE^{p=5\%}$. This discrepancy arises because, in $SPO_+^{p=5\%}$, the optimization problem is solved for $2\hat{y} - y$. Solving for $2\hat{y} - y$ is more challenging and time-consuming compared to solving for $\hat{y}$, as done in $SCE^{p=5\%}$. This is due to the difference in scale between the true cost ($y$) and the predicted cost ($\hat{y}$) We point that this pattern is also visible in Figure 9. The computational burden of $SPO_+^{p=5\%}$ becomes especially pronounced for the larger problem instances. For these instances, solving the optimization problem with $2\hat{y} - y$ often results in timeouts, meaning Gurobi returns an approximate solution instead of the exact one. This is the reason why $SPO_+^{p=5\%}$ exhibits relatively higher regret than $SCE^{p=5\%}$ for these problems.

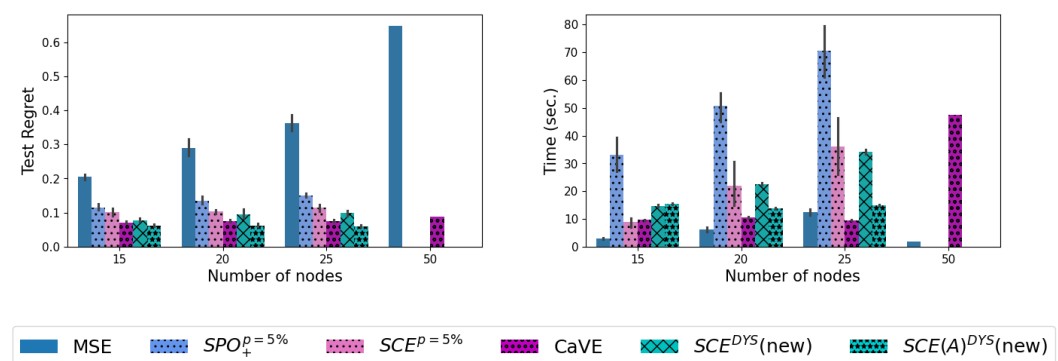

Figure 10: Comparative analysis on larger TSP instances.

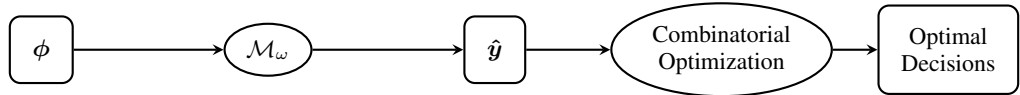

Figure 11: Schematic diagram of a predict-then-optimize (PtO) problem.

In contrast, for $SCE^{p=5\%}$, timeouts never occurred, and it exhibits lower training times compared to $SPO_+^{p=5\%}$. For TSP with 15 nodes, $SCE^{p=5\%}$ outperforms DYS-Net in terms of training time. However, as problem size increases, DYS-Net demonstrates better scalability, whereas solving the optimization problem even for 5% of the training instances becomes significantly time-intensive in $SCE^{p=5\%}$.

In terms of regret, DYS-Net demonstrates a significant advantage with much lower regret compared to other methods. This underscores the advantage of minimizing $\mathcal{L}_{SCE}$ using DYS-Net, as it not only delivers lower regret but also scales more effectively for larger problems.

## H   PREDICT-THEN-OPTIMIZE PROBLEM DESCRIPTION

We consider predicting parameters in the objective function of an LP. These kinds of problems can be framed as *predict-then-optimize* (PtO) problems consisting of a prediction stage followed by an optimization stage, as illustrated in Figure 11. In the prediction stage, an ML model $\mathcal{M}_\omega$ (with trainable parameters $\omega$) is used to predict unknown parameters using features, $\phi$, that are correlated to the parameter. During the optimization stage, the problem is solved with the predicted parameters. An offline dataset of past observations is available for training $\mathcal{M}_\omega$.

It is important to distinguish datasets based on whether the true parameters, $\boldsymbol{y}$, are observed and included in the dataset. In some applications, the true parameters, $\boldsymbol{y}$, may not be directly observable, and only the solutions, $\boldsymbol{v}^\star(\boldsymbol{y})$, are observed. While $\boldsymbol{v}^\star(\boldsymbol{y})$ can be computed if $\boldsymbol{y}$ is known, the reverse is not true, since solving the inverse optimization problem is a separate research area.

Whether $\boldsymbol{y}$ is observed or not is important because in order to compute $Regret$ (equation 2), we need the true parameter $\boldsymbol{y}$. Most of the benchmarks in PtO problems assume that $\boldsymbol{y}$ is observed in the past observation. In this case the training data can be expressed as $\{(\phi_i, \boldsymbol{y}_i, \boldsymbol{v}^\star(\boldsymbol{y}_i))\}_{i=1}^N$ and the empirical regret, $\frac{1}{N}\sum_{i=1}^N Regret(\boldsymbol{v}^*(\mathcal{M}_\omega(\phi_i)), \boldsymbol{y}_i)$, can be computed. In most PtO benchmark problems it is assumed that the true $\boldsymbol{y}$ is observed in the training data (Mandi et al., 2024; Tang & Khalil, 2023). However, if the true cost $\boldsymbol{y}$ is not observed in the training data, empirical regret cannot be computed. Instead, some other loss must be considered. For instance, McKenzie et al. (2024) consider squared decision errors (SqDE) between $\boldsymbol{v}^\star(\boldsymbol{y})$ and $\boldsymbol{v}^\star(\hat{\boldsymbol{y}})$, i.e., $SqDE = ||\boldsymbol{v}^\star(\boldsymbol{y}) - \boldsymbol{v}^\star(\hat{\boldsymbol{y}})||^2$.

---

**Algorithm 1** Gradient-descent with Smoothing

---

1: Initialize $\boldsymbol{\omega}$.
2: **for** each epoch **do**
3:    **for** each instance $(\boldsymbol{\phi}, \boldsymbol{y}, \boldsymbol{v}^\star(\boldsymbol{y}))$ **do**
4:       $\hat{\boldsymbol{y}} = \mathcal{M}_\omega(\boldsymbol{\phi})$
5:       Obtain $\boldsymbol{v}^\star(\hat{\boldsymbol{y}})$ by solving a 'smoothed' optimization
6:       $Regret(\boldsymbol{v}, \boldsymbol{y}) = \boldsymbol{y}^\top \boldsymbol{v}^\star(\hat{\boldsymbol{y}}) - \boldsymbol{y}^\top \boldsymbol{v}^\star(\boldsymbol{y})$
7:       $\boldsymbol{\omega} \leftarrow \boldsymbol{\omega} - \alpha \frac{dRegret(\boldsymbol{v},\boldsymbol{y})}{d\hat{\boldsymbol{y}}} \frac{d\hat{\boldsymbol{y}}}{d\boldsymbol{\omega}}$
8:    **end for**
9: **end for**

---

**Algorithm 2** Gradient-descent with Surrogate Losses

---

1: **for** each epoch **do**
2:    **for** each instance $(\boldsymbol{\phi}, \boldsymbol{y}, \boldsymbol{v}^\star(\boldsymbol{y}))$ **do**
3:       $\hat{\boldsymbol{y}} = \mathcal{M}_\omega(\boldsymbol{\phi})$
4:       Compute $\tilde{\boldsymbol{y}}$
5:       Obtain $\boldsymbol{v}^\star(\tilde{\boldsymbol{y}})$ by solving the original optimization
6:       Compute the surrogate loss $\mathcal{L}$ and $\nabla$
7:       $\boldsymbol{\omega} \leftarrow \boldsymbol{\omega} - \alpha \nabla \frac{d\hat{\boldsymbol{y}}}{d\boldsymbol{\omega}}$
8:    **end for**
9: **end for**

---

**Algorithm 3** Gradient-descent when Surrogate Losses are minimized using Smoothed Solver

---

1: **for** each epoch **do**
2:    **for** each instance $(\boldsymbol{\phi}, \boldsymbol{y}, \boldsymbol{v}^\star(\boldsymbol{y}))$ **do**
3:       $\hat{\boldsymbol{y}} = \mathcal{M}_\omega(\boldsymbol{\phi})$
4:       Compute $\tilde{\boldsymbol{y}}$
5:       Obtain $\boldsymbol{v}^\star(\tilde{\boldsymbol{y}})$ by solving a 'smoothed' optimization
6:       Compute the surrogate loss $\mathcal{L}$
7:       $\boldsymbol{\omega} \leftarrow \boldsymbol{\omega} - \alpha \frac{d\mathcal{L}}{d\hat{\boldsymbol{y}}} \frac{d\hat{\boldsymbol{y}}}{d\boldsymbol{\omega}}$
8:    **end for**
9: **end for**

---

## I    DIFFERENT APPROACHES TO DECISION-FOCUSED LEARNING

In PtO problems, the empirical regret can be calculated if the cost, $\boldsymbol{y}$, is observed in the training instances. However, just because it can be calculated does not mean it can be minimized using gradient descent. Figure 12 illustrates the impact of integrating the optimization block into the training loop of neural networks. The key challenge is that to directly minimize $Regret$, it must be backpropagated through the optimization problem. However, for a combinatorial problem $\boldsymbol{v}^\star(\hat{\boldsymbol{y}})$ does not change smoothly with $\hat{\boldsymbol{y}}$, so the gradient, $\frac{d\boldsymbol{v}^\star(\hat{\boldsymbol{y}})}{d\hat{\boldsymbol{y}}}$, is either zero or does not exist.

**Differentiable Optimization by Smoothing.** 'Differentiable Optimization by Smoothing' is one approach to to circumvent this challenge. The aim of differentiable optimization is to represent an optimization problem as a differentiable mapping from its parameters to its solution. Since for a COP, this mapping is **not** differentiable, one prominent research direction in DFL involves smoothing the combinatorial optimization problem into a differentiable optimization problem. We particularly focus on smoothing by regularization. There exists another from of smoothing—smoothing by perturbation, as proposed by Pogančić et al. (2020); Blondel et al. (2020); Niepert et al. (2021); Sahoo et al. (2023). In this work, we focus on optimization problems with linear objective functions such as LPs and ILPs. For an LP, the solution will always lie in one of the vertices of the LP simplex. So, the LP solution remains unchanged as long as the cost vector changes while staying within the corresponding normal

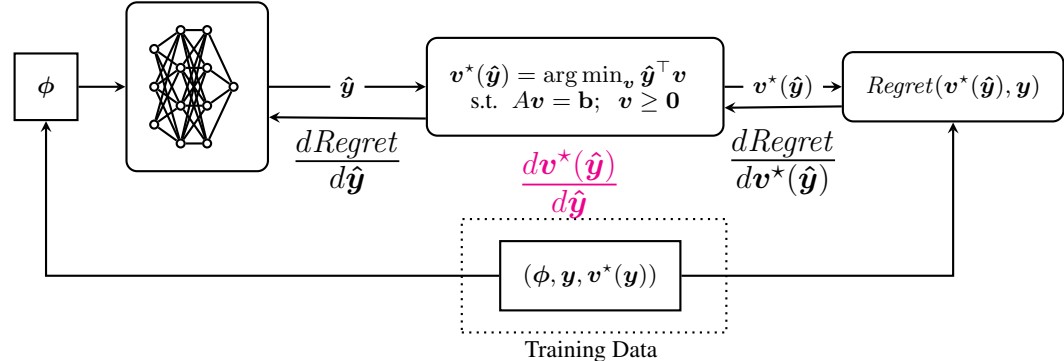

Figure 12: Decision-focused learning training loop.

cone (Boyd & Vandenberghe, 2004). However, the solution will suddenly switch to a different vertex if the cost vector slightly moves outside the normal cone, as illustrated in Figure 1a. Because the solution abruptly jumps between the vertices, the LP solution is not a differentiable function of the cost vector.

As explained in Section 3.1, approaches under this category replace the original optimization problem with a 'smoothed' version of the optimization problem, in which the solution can be expressed as a differentiable mapping of the parameter. For instance, if the original problem is an LP, it can be replaced with a QP by adding a quadratic regularizer to the objective of the LP. In this QP, the solution, $\boldsymbol{v}^\star(\boldsymbol{y})$, can be represented as a differentiable function of the parameter $\boldsymbol{y}$. When the problem is an ILP, first LP, resulting from continuous relaxation is considered and then it is smoothed by adding quadratic regularizer. Algorithm 1 explains this approach. DYS-Net (McKenzie et al., 2024) provides an approximate solution to the quadratically regularized LP problem, where the computations are designed to be executed as standard neural network operations, enabling back-propagation through it. To summarize, approaches in this category follow the training loop in Figure 12 but only after 'smoothing' the optimization problem.

**Surrogate Losses for DFL.** The primary goal of DFL is to minimize $Regret$. However, as explained earlier, $Regret$ cannot be minimized directly due to its non-differentiability. Techniques involving surrogate losses aim to address this challenge by identifying suitable surrogate loss functions and computing gradients or subgradients of these surrogate losses for optimization. Figure 13 depicts the training loop of DFL using surrogate loss functions. In this approach, $Regret(\boldsymbol{v}^\star(\hat{\boldsymbol{y}}), \boldsymbol{y})$ is not explicitly computed. Instead, after predicting $\hat{\boldsymbol{y}}$, a new cost vector $\tilde{\boldsymbol{y}}$ is generated based on $\hat{\boldsymbol{y}}$ and $\boldsymbol{y}$, and the optimization problem is solved using this $\tilde{\boldsymbol{y}}$. Subsequently, a surrogate loss is computed, using $\boldsymbol{v}^\star(\tilde{\boldsymbol{y}})$ and $\boldsymbol{v}^\star(\boldsymbol{y})$, and its gradient,$\nabla$ (shown in pink) , is used for backpropagation. We have explained this in terms of pseudocode using Algorithm 2. For example, in the case of $\mathcal{L}_{SPO+}$, $\tilde{\boldsymbol{y}} = 2\hat{\boldsymbol{y}} - \boldsymbol{y}$. As shown in Equation 4 $\mathcal{L}_{SPO+} = (2\hat{\boldsymbol{y}} - \boldsymbol{y})^\top \boldsymbol{v}^\star(\boldsymbol{y}) - (2\hat{\boldsymbol{y}} - \boldsymbol{y})^\top \boldsymbol{v}^\star(2\hat{\boldsymbol{y}} - \boldsymbol{y})$ Then the gradient used for backpropagation is $\nabla = 2(\boldsymbol{v}^\star(\boldsymbol{y}) - \boldsymbol{v}^\star(2\hat{\boldsymbol{y}} - \boldsymbol{y}))$. On the other hand, in the case of $\mathcal{L}_{SCE}$, $\tilde{\boldsymbol{y}} = \hat{\boldsymbol{y}}$ and $\mathcal{L}_{SCE} = \hat{\boldsymbol{y}}^\top (\boldsymbol{v}^\star(\boldsymbol{y}) - \boldsymbol{v}^\star(\hat{\boldsymbol{y}})) + \boldsymbol{y}^\top (\boldsymbol{v}^\star(\hat{\boldsymbol{y}}) - \boldsymbol{v}^\star(\boldsymbol{y}))$. So, in this case, the gradient for backpropagation is $\nabla = (\boldsymbol{v}^\star(\boldsymbol{y}) - \boldsymbol{v}^\star(\hat{\boldsymbol{y}}))$.

**Combining Surrogate Losses with Differentiable Optimization.** The core idea proposed in this paper is to combine these two approaches. Specifically, the original optimization problem in Figure 13 is replaced with a smoothed version, **allowing direct backpropagation through the smoothed problem instead of using** $\nabla$. We emphasis that this changes the gradient of the surrogate losses. Instead of Equation 5 and Equation 8, Equation 11 Equation 12 will be used in this case for backprogating $\mathcal{L}_{SPO+}$ and $\mathcal{L}_{SCE}$ respectively. We explain this in Algorithm 3.

## J    GENERALIZATION BOUNDS FOR NCE LOSS

We show generalization bounds for SCE loss similar to the bounds shown for true regret by El Balghiti et al. (2019). For notational brevity, we first define SCE in terms of the predicted and true parameters,

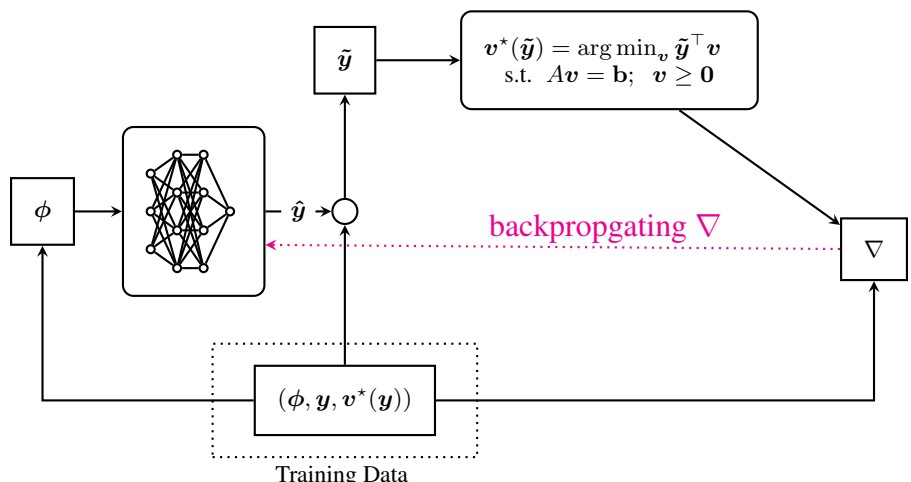

Figure 13: Decision-focused learning using surrogate loss functions.

i.e.,

$$l_{SCE}(\hat{\boldsymbol{y}}, \boldsymbol{y}) = \mathcal{L}_{SCE}(\boldsymbol{v}^\star(\hat{\boldsymbol{y}}), \boldsymbol{y}) = (\hat{\boldsymbol{y}} - \boldsymbol{y})^\top (\boldsymbol{v}^\star - \boldsymbol{v}^\star(\hat{\boldsymbol{y}}))$$

where $\hat{\boldsymbol{y}} = \mathcal{M}(\boldsymbol{\phi})$ is the predicted cost using the predcitive model $\mathcal{M}$. We can also define

$$R_{SCE}(\mathcal{M}) = \mathbb{E}[l_{SCE}(\mathcal{M}(\boldsymbol{\phi}), y)] \text{ and } \widehat{R}_{SCE}(\mathcal{M}) = \frac{1}{N} \sum_{i=1}^N l_{SCE}(\mathcal{M}(\boldsymbol{\phi}_i), y_i)$$

as the true and empirical risk for a given sample $\{(\boldsymbol{\phi}_i, \boldsymbol{y}_i)\}_{i=1}^N$ for SCE loss, respectively.

In order to show generalization bounds for SCE loss, we need to define the Rademacher complexity of a set of functions $\mathcal{H}$ with $l_{SCE}$. The sample Rademacher complexity for a given sample $\{(\boldsymbol{\phi}_i, \boldsymbol{y}_i)\}_{i=1}^N$ is given by

$$\widehat{\mathfrak{R}}_{SCE}^N(\mathcal{H}) = \mathbb{E}_\sigma \left[ \sup_{\mathcal{M} \in \mathcal{H}} \frac{1}{N} \sum_{i=1}^N \sigma_i l_{SCE}(\mathcal{M}(\boldsymbol{\phi}_i), \boldsymbol{y}_i) \right]$$

where $\sigma_1, \sigma_2, \ldots, \sigma_N$ are i.i.d. random variables with $\mathbb{P}(\sigma_i = 1) = 1/2$ and $\mathbb{P}(\sigma_i = -1) = 1/2$ for $i = 1, 2, \ldots, N$. The expected Rademacher complexity is defined as $\mathfrak{R}_{SCE}^N(\mathcal{H}) = \mathbb{E}[\widehat{\mathfrak{R}}_{SCE}^N(\mathcal{H})]$ where the expectation is with respect to the i.i.d. samples of size $N$ from the true distribution.

Assume that the set of all feasible solutions $\mathcal{F} = \{\boldsymbol{v} : A\boldsymbol{v} = \mathbf{b}; \ \boldsymbol{v} \geq \mathbf{0}\}$ is bounded, i.e., there exists $D$ such that $D = max_{\boldsymbol{v}, \boldsymbol{v}' \in \mathcal{F}} \|\boldsymbol{v} - \boldsymbol{v}'\|$. Also assume that the set of all cost vectors is $\mathcal{Y}$ such that $\mathcal{Y} \subseteq \{\boldsymbol{y} : \|\boldsymbol{y}\| \leq 1\}$. Note that, since we consider linear objective functions, this assumption is not restrictive, as $\boldsymbol{y}'$ with $\|\boldsymbol{y}'\| > 1$ can be replaced by $\boldsymbol{y} = \boldsymbol{y}'/\|\boldsymbol{y}'\|$ without changing the optimal solution and ensuring $\|\boldsymbol{y}\| = 1$.

The following proposition shows the generalization bound for SCE loss.

**Proposition 2.** *Let $\mathcal{H}$ be a set of functions from the set of all features to $\{\boldsymbol{y} : \|\boldsymbol{y}\| \leq 1\}$. Then for any $\delta > 0$,*

$$R_{SCE}(\mathcal{M}) - \widehat{R}_{SCE}(\mathcal{M}) \leq 2\mathfrak{R}_{SCE}^N(\mathcal{H}) + 2D\sqrt{\frac{\log(1/\delta)}{2N}} \tag{21}$$

*holds for all $\mathcal{M} \in \mathcal{H}$ with probability at least $1 - \delta$ for the sample $\{(\boldsymbol{\phi}_i, \boldsymbol{y}_i)\}_{i=1}^N$ from the joint distribution of features and parameters. If $\widehat{\mathcal{M}}_n$ is a minimizer of the emprical risk $\widehat{R}_{SCE}$, then the inequality*

$$R_{SCE}(\widehat{\mathcal{M}}_n) - \min_{\mathcal{M} \in \mathcal{H}} R_{SCE}(\mathcal{M}) \leq 2\mathfrak{R}_{SCE}^N(\mathcal{H}) + 4D\sqrt{\frac{\log(2/\delta)}{2N}} \tag{22}$$

*also holds probability at least $1 - \delta$.*

*Proof.* The SCE loss is bounded for all $\boldsymbol{y}, \boldsymbol{y}' \in \mathcal{Y}$ since $l_{SCE}(\hat{\boldsymbol{y}}, \boldsymbol{y}) = (\hat{\boldsymbol{y}} - \boldsymbol{y})^\top (\boldsymbol{v}^\star - \boldsymbol{v}^\star(\hat{\boldsymbol{y}})) \leq \|\hat{\boldsymbol{y}} - \boldsymbol{y}\| \|\boldsymbol{v}^\star - \boldsymbol{v}^\star(\hat{\boldsymbol{y}})\| \leq 2D$ where the first inequality is due to Cauchy–Schwarz and the second inequality is due to our assumptions on the hypothesis class and the feasible region. Then, inequality 21 follows directly from the classical generalization bound as shown in Bartlett & Mendelson (2002).

The extension of inequality 21 to inequality 22 is shown in the proof of Corollary 1 in El Balghiti et al. (2019) using Hoeffding's inequality. $\square$

## K    Detailed Description of The Experimental Setup

In this section, we first describe the optimization problems along with their formulations, followed by details of the data generation process and the ML models.

### K.1    Description of the Optimization Problems

**Shortest Path Problem.**    It is a shortest path problem on a $k \times k$ grid, with the objective of going from the southwest corner of the grid to the northeast corner where the edges can go either north or east. This grid consists of $k^2$ nodes and $2 \times k \times (k-1)$ edges. Let, $y_{ij}$ is the cost of going from node $i$ to node $j$ and the decision variable $v_{ij}$ takes the value 1 if and only if the edge from node $i$ to node $j$ is traversed. Then, the shortest path problem from going to node $s$ to node $t$ can be formulated as an LP problem in the following form:

$$\min_{v_{ij}} \sum_{(i,j) \in \mathcal{E}} y_{ij} v_{ij} \tag{23a}$$

s.t.

$$\sum_{(i,j) \in \mathcal{E}} v_{ij} - \sum_{(k,i) \in \mathcal{E}} v_{ki} = \begin{cases} 1 & \text{if } i = s \\ -1 & \text{if } i = t \\ 0 & \text{otherwise} \end{cases} \tag{23b}$$

$$v_{ij} \in \mathbb{R}^+ \tag{23c}$$

**Knapsack Problem.**    In a knapsack problem the goal of the optimization problem is to choose a maximal value subset from a given set of items, subject to some capacity constraints. Let the set contains $N_{items}$ number of items and the value of each item is $y_i$. The solution must satisfy capacity constraints in multiple dimensions. Let $C_j$ is the capacity in dimension $j$ and $w_{(i,j)}$ is the weight of item $i$ in dimension $j$. This optimization can be modeled as an integer linear programming (ILP)m as follows:

$$\min_{v_i} \sum_{i=1}^{N_{items}} (-y_i) v_i \tag{24a}$$

s.t.

$$\sum_{i=1}^{N_{items}} w_{(i,j)} v_i \leq C_j \; ; \; \forall j \tag{24b}$$

$$v_i \in \{0, 1\} \tag{24c}$$

The Top-K selection can be viewed as a special case of the knapsack problem. In the Top-K, there is only one dimension and the weight of each item is 1 and the capacity, $C = K$.

**Traveling Salesperson Problem.**    Given a topological graph of $N_{nodes}$, the objective of the traveling salesperson problem (TSP) is to find the shortest possible tour that visits every node exactly once. We formulate the TSP as an mixed integer linear programming (MILP) following the Miller-Tucker-Zemlin (MTZ) formulation so that we can solve the relaxed LP, with quadratic reglarizer, using DYS-Net. Let, $y_{ij}$ is the cost of going from node $i$ to node $j$ and the decision variable $v_{ij}$ takes the value 1 if and only if the salesperson traverse from node $i$ to node $j$. Then the MTZ formulation is

the following:

$$\min_{v_{ij}} \sum_{i=1}^{N_{nodes}} \sum_{j=1}^{N_{nodes}} y_{ij} v_{ij} \tag{25a}$$

s.t.

$$\sum_{j=1}^{N_{nodes}} v_{ij} = 1 \quad \forall i \tag{25b}$$

$$\sum_{i=1}^{N_{nodes}} v_{ij} = 1 \quad \forall j \tag{25c}$$

$$u_j - u_i \geq 1 + N_{nodes}(v_{ij} - 1); \quad 2 \leq i, j \leq N_{nodes} \tag{25d}$$

$$v_{ij} \in \{0, 1\}, \ u_i \in \mathbb{R}^+ \tag{25e}$$

Note, for other techniques we can solve the TSP using Dantzig–Fulkerson–Johnson (DFJ) formulation, which is faster.

## K.2 DESCRIPTION OF THE DATA GENERATION PROCESS

We use PyEPO library (Tang & Khalil, 2023) to generate training, validation and test datasets. Each dataset consists of $\{(\phi_i, y_i)\}_{i=1}^N$, which are generated synthetically. The feature vectors are sampled from a multivariate Gaussian distribution with zero mean and unit variance, i.e., $\phi_i \sim \mathbf{N}(0, I_p)$, where p is the dimension of $\phi_i$. To generate the cost vector, first a matrix $B \in \mathbb{R}^{K \times p}$ is generated, which represents the true underlying model, unknown to the modeler. Each element in the cost vector $y_{i,j}$ is then generated according to the following formula:

$$\boldsymbol{y}_{ij} = \left[ \frac{1}{3.5^{\text{Deg}}} \left( \frac{1}{\sqrt{p}} (B\phi_i) + 3 \right)^{\text{Deg}} + 1 \right] \xi_i^j \tag{26}$$

The *Deg* is 'model misspecification' parameter. This is because a linear model is used as a predictive model in the experiment and a higher value of *Deg* indicates the predictive model deviates more from the true underlying model and larger the prediction errors. $\xi_i^j$ is a multiplicative noise term sampled randomly from the uniform distribution $\xi_i^j \sim U[1 - w, 1 + w]$. $w$ is a noise-half width parameter, which is less than 1. Higher values of $w$ indicate a greater degree of noise perturbation. We set *Deg* to 6 and $w$ to 0.5 in all our experiments.

## L IMPLEMENTATION OF DYS

We adopt the implementation by McKenzie et al. (2024) to implement DYS-Net [1]. DYS-Net includes a few hyperparameters: $\mu$, controls the strength of smoothing; scaling parameter $\alpha \in (0, 2/\mu)$; number of time Equation DYS is iterated. (For detailed explanations of these parameters, please refer to the original papers.) In practice, we set $\mu \approx 0$ and the number of iterations to 200. Each iteration is implemented as a multi-layer perceptron (MLP), making the implementation computationally efficient. We tune the parameter $\alpha$ on a validation set. We also tried slowly decreasing $\alpha$ across the iterations. However, reduction of $\alpha$ has little effect on the result. Notably, this implementation does not require pretraining, as DYS-Net contains trainable parameters.

**Ablation Analysis.** We present how the performance of DYS-Net varies with different hyperparameter settings in Table 2. The results reported are based on the test set. However, we do not evaluate performance across varying values of $\alpha$ and $\mu$, as these were determined through hyperparameter tuning on the validation set. We observe, setting the values of $\mu$ low works well for most problems. We choose $\alpha$ from the set $\{0.001, 0.01, 0.1\}$. This setup has proven effective across different problems.

---

[1] https://github.com/mines-opt-ml/fpo-dys

| Model | $\alpha$ | $\mu$ | Normalized relative regret | |
|---|---|---|---|---|
| | | | Average | Sd |
| $SCE^{DYS}$ | 0.1 | 0.001 | 0.075 | 0.016 |
| $SCE^{DYS}$ | 0.01 | 0.001 | 0.088 | 0.014 |
| $SCE^{DYS}$ | 0.01 | 1. | 0.104 | 0.014 |
| $SCE^{DYS}$ | 0.01 | 10. | 0.119 | 0.021 |
| $SCE^{DYS}$ | 1. | 0.001 | 0.190 | 0.044 |

Table 2: Ablation of $SCE^{DYS}$ on TSP-8 problem instances.

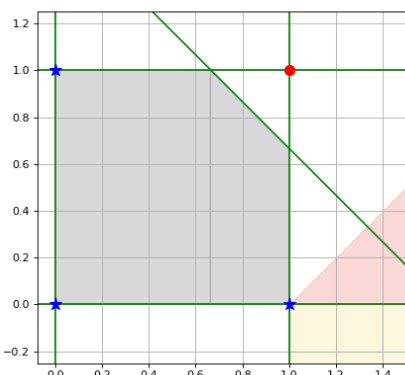

Figure 14: A numerical illustration to show why the Cave approach fails in the Knapsack problems.

## M  SHORTCOMING OF CAVE ON THE KNAPSACK PROBLEM

Consider the two-dimensional knapsack example in Figure 14. The capacity constraint is given as $3v_1 + 3v_2 \leq 5$ . If the objective vector $\boldsymbol{y}$ lies within the union of the yellow and red cones, then the feasible solution $(1,0)$ is optimal for the problem with the integrality constraint. So, the true normal cone is the union of the yellow and red cones. Note that the constraint $3v_1 + 3v_2 \leq 5$ is not active, although it plays a key role in choosing the solution; in the absence of this constraint, the solution would be $(1,1)$. In this case, the only active constraints are $v_1 = 1$ and $v_2 = 0$. As the CaVE approach stores only these two constraints, the yellow cone is considered as the optimality cone. This example shows that the mismatch between the cone of optimality of the integer knapsack and its relaxation can be non-trivial (the red cone in Figure 14). This attributes to the poor performance of the CaVE approach in the Knapsack problem.

## N  LEARNING CURVES

Figure 15, Figure 16 and 17 illustrate how the regret on the validation dataset evolves for different losses as training progresses for the KP, SP and TSP problem instances when the DYS-Net is used as a solver. It shows that in general, training with $\mathcal{L}_{SCE}$ results in lower regret than training with $Regret$ or $SqDE$.

## O  ADDITIONAL EXPERIMENTS

### O.1  REGRET VS. SURROGATE LOSS WITH QP SMOOTHING

We use *CvxpyLayer* (Agrawal et al., 2019) to solve and differentiate through the smooth optimization problem after adding the quadratic regularizer. The column $MSE$ corresponds to ML models trained

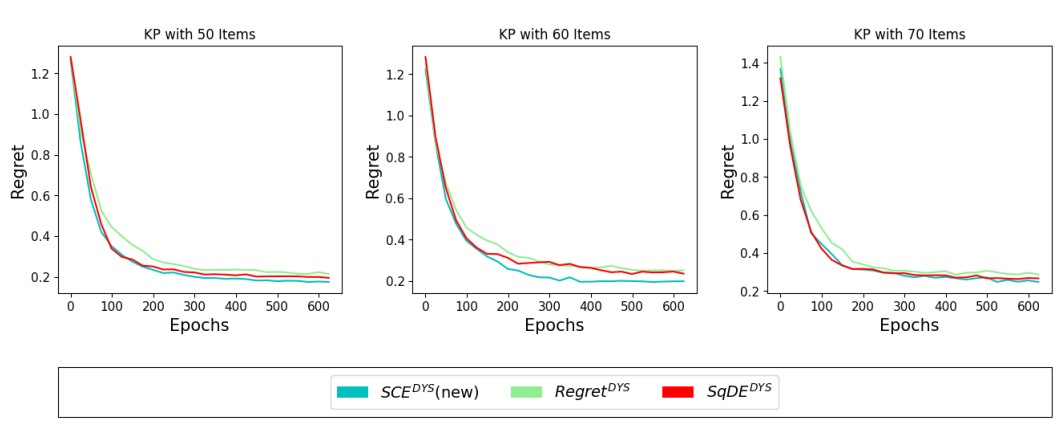

Figure 15: Progression of Training for the 3 KP problems.

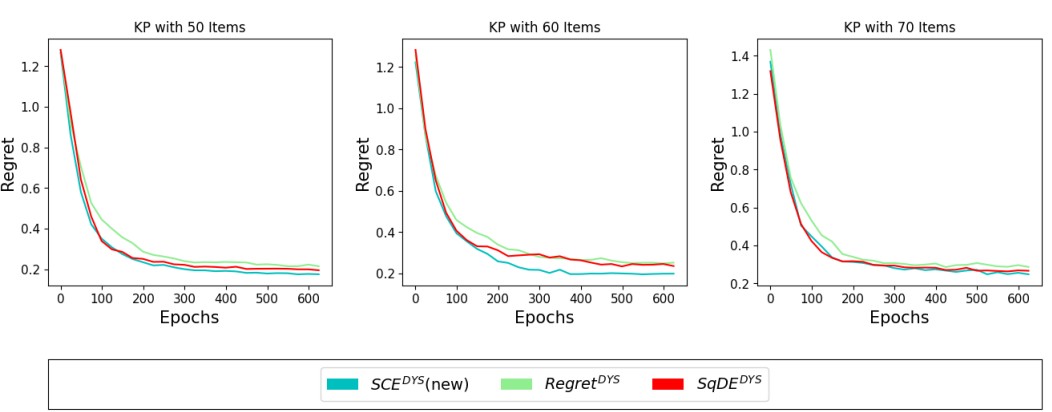

Figure 16: Progression of Training for the 3 SP problems.

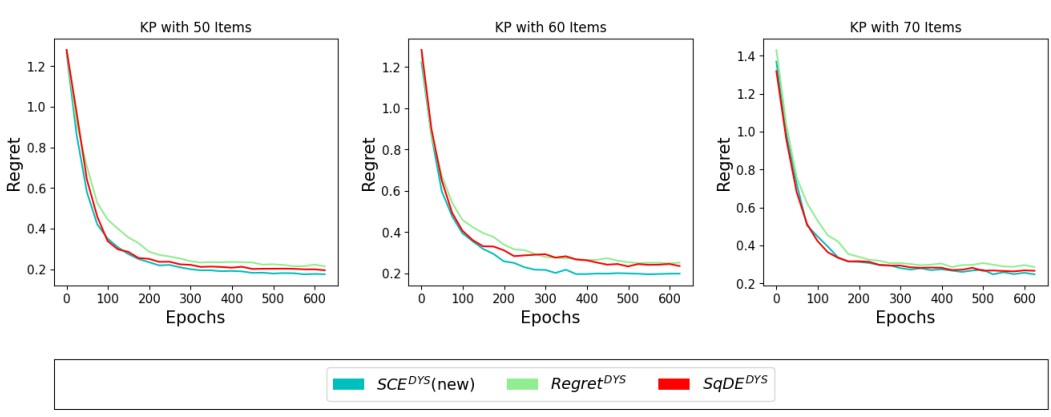

Figure 17: Progression of Training for the 3 TSP problems.

Table 3: Normalized relative regret on test data for four optimization problems. We mention the number of resources, the size of the grid, the number of items and the number of nodes for the Top-K, shortest path, knapsack and TSP problems respectively in the parenthesis.

| | | Combinatorial | | | CvxpyLayer | | |
|---|---|---|---|---|---|---|---|
| | MSE | PFY | $\mathcal{L}_{SPO+}$ | $\mathcal{L}_{SCE}$ | Regret | $\mathcal{L}_{SPO+}$ | $\mathcal{L}_{SCE}$ |
| Top-K (50) | 1.614 ±0.874 | **0.051** ±0.006 | **0.051** ±0.006 | **0.051** ±0.006 | 0.246 ±0.439 | **0.051** ±0.006 | **0.051** ±0.006 |
| Top-K (80) | 1.622 ±0.896 | 0.018 ±0.001 | **0.018** ±0.001 | **0.018** ±0.001 | 0.419 ±0.896 | **0.018** ±0.001 | **0.018** ±0.001 |
| Top-K (100) | 1.623 ±0.9 | **0.013** ±0.001 | **0.013** ±0.001 | **0.013** ±0.001 | 0.214 ±0.45 | **0.013** ±0.001 | **0.013** ±0.001 |
| SP (5 × 5) | 0.45 ±0.124 | 0.328 ±0.037 | **0.302** ±0.042 | 0.431 ±0.06 | 0.339 ±0.035 | **0.303** ±0.044 | **0.303** ±0.032 |
| SP (8 × 8) | 0.539 ±0.064 | **0.425** ±0.048 | 0.447 ±0.038 | 0.632 ±0.082 | 0.486 ±0.041 | 0.454 ±0.031 | 0.445 ±0.036 |
| SP (10 × 10) | 0.492 ±0.113 | 0.462 ±0.118 | 0.443 ±0.103 | 0.626 ±0.165 | 0.745 ±0.174 | 0.442 ±0.105 | **0.424** ±0.111 |
| KP (10) | 0.129 ±0.051 | **0.098** ±0.049 | 0.101 ±0.034 | 0.163 ±0.009 | 0.197 ±0.047 | 0.11 ±0.032 | 0.104 ±0.044 |
| KP (20) | 0.174 ±0.037 | **0.128** ±0.035 | 0.134 ±0.037 | 0.16 ±0.035 | 0.222 ±0.075 | 0.139 ±0.027 | **0.129** ±0.029 |
| KP (40) | 0.176 ±0.019 | **0.149** ±0.011 | 0.142 ±0.008 | 0.17 ±0.011 | 0.217 ±0.025 | 0.153 ±0.008 | **0.146** ±0.009 |
| TSP (5) | 0.101 ±0.036 | 0.079 ±0.032 | **0.067** ±0.028 | 0.152 ±0.05 | 0.095 ±0.029 | 0.078 ±0.027 | 0.073 ±0.026 |
| TSP (6) | 0.111 ±0.021 | **0.06** ±0.015 | **0.059** ±0.014 | 0.161 ±0.071 | 0.069 ±0.009 | 0.081 ±0.01 | **0.059** ±0.006 |
| TSP (8) | 0.12 ±0.008 | 0.072 ±0.011 | 0.071 ±0.013 | 0.117 ±0.021 | 0.081 ±0.01 | 0.095 ±0.011 | **0.065** ±0.012 |

with the MSE loss between $y$ and $\hat{y}$. As this approach does not consider the optimization problem during training, we anticipate it would have higher regret than the DFL approaches. Implementation of perturbed Fenchel-Young (PFY) (Berthet et al., 2020), $\mathcal{L}_{SPO+}$ and $\mathcal{L}_{SCE}$ using *combinatorial solvers* serve as three DFL benchmarks. We choose $\mathcal{L}_{SPO+}$ and PFY, as they are best performing DFL methods across various optimization problems (Mandi et al., 2024; Tang & Khalil, 2023). When $\mathcal{L}_{SPO+}$ and $\mathcal{L}_{SCE}$ are minimized using combinatorial solvers, Eq. 5 and Eq. 8 are used for gradient backpropagation. The three columns under *CvxpyLayer* show regret when the losses are backpropagated through the smoothed QP problem using *CvxpyLayer*. *Regret* appears only under *CvxpyLayer*, because it can only be minimized after QP smoothing. This paper is the first to test the last two approaches, which combine differential smoothing and surrogate losses.

For the Top-K problem, all DFL approaches have exact same regret. We explain this behaviour in the appendix. Next, we highlight that in all cases, *minimizing $\mathcal{L}_{SPO+}$ or $\mathcal{L}_{SCE}$ results in lower test regret than minimizing Regret using CvxpyLayer*, which corroborates the main proposal we made in this paper. Across all experiments, we observe that minimizing $\mathcal{L}_{SCE}$ using *CvxpyLayer* yields regret similar to $\mathcal{L}_{SPO+}$ and PFY, which use combinatorial solvers. This shows that minimizing $\mathcal{L}_{SCE}$ using *CvxpyLayer* can compete with the state-of-the-art in DFL. Moreover, $\mathcal{L}_{SPO+}$ produce

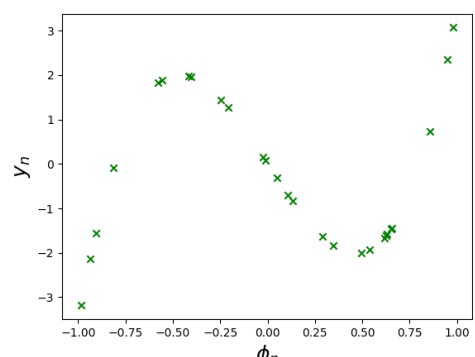

Figure 18: Relationship between $y_n$ and $\phi_n$ in the Top-K experiment.

lower regret, when a combinatorial solver is used, whereas $\mathcal{L}_{SCE}$ performs better with *CvxpyLayer*. This opens up an interesting side observation— *Eq. 5 ($\mathcal{L}_{SPO+}$ ) provide a better subgradient than Eq. 8 ($\mathcal{L}_{SCE}$). However, when one can differentiate through the optimization, $\mathcal{L}_{SCE}$ (Eq. 12) has a better gradient than $\mathcal{L}_{SPO+}$ (Eq. 11).*

### O.2  EXPLANATION OF THE TOP-K DATASET

In the Top-K experiments, the relationship between $y_n$ and $\phi_n$ is illustrated in Figure 18. All DFL models learn a mapping with a positive slope. As a result, each model selects the Top-1 element as the one with the highest value of $\phi_n$, leading to identical accuracy across all DFL models. In contrast, models trained with MSE loss fail to learn a positive slope, resulting in significantly higher regret.

