# OpenReview forum: "Combining Analytical Smoothing with Surrogate Losses for Improved Decision-Focused Learning"
_ICLR.cc/2025/Conference — Submitted to ICLR 2025_

### Official Review · Reviewer_e6tg · 2024-11-01

**Soundness:** 2
**Presentation:** 2
**Contribution:** 2
**Rating:** 3
**Confidence:** 4

**Summary:**

This paper considers a new training method for decision-focused learning, also known as end-to-end learning or joint predict-then-optimize. They explain two existing methods to approach these problems: surrogate loss approaches such as the smart predict the optimize (SPO) and contrastive loss as well as smoothing methods. The authors propose to combine the two methods, minimizing a surrogate loss with a smoothed solver. They run experiments on some optimization tasks like shortest path, knapsack and TSP.

**Strengths:**

The ideas are simple and can be easily implemented by practitioners. You present a good outline of past work and background on existing methods. The experimental results are promising as well and show improvements of your proposed method against some of the main existing literature.

**Weaknesses:**

The exposition and contribution is not very clearly expressed in the paper. As a starting point, it is difficult to find where your proposed method is written. I'm guessing your proposed approach is to solve an empirical version of problem (2). The actual empirical problem is never introduced. I also assume that, given features $x$, you would construct a model to make predictions $f(x)$ of $y$. This is also not explained well. Finally, you solve the empirical problem by gradient descent using (12). All of this needs to be made clear and cohesive. I would suggest a section to explain the full problem and proposed method.

For example, the statement "smoothing addresses the non-differentiability at the transition points, but the derivative dv⋆( ˆy)
d ˆy still remains zero far from these points" is not well-supported. The illustration in Figure 1b that the authors point to does not give adequate support for these claims as a single low-dimensional example does not generalize. Similarly, your subsection "A deep dive into the gradient landscape" does not adequately explain the behavior of your approach. Why should it generalize to more complex problems? Please provide some theoretical analysis on the gradient behavior in higher dimension. Alternatively, you can also provide computational results on gradient behavior for various optimization tasks and higher dimension.

I'm a little confused why section 5 exists. This is only explaining the methods used in a different paper to implement your proposed method. It is not a contribution of your work, as far as I can tell. It may be better to place this discussion in the appendix, if at all. Moreover, the work in [1] may be a good additional reference as it seems to be a more generalized version of the work you cite.

Finally, for the experiments, please give more details on the problems addressed. For example, provide information about:
    1. Problem sizes for each experiment
    2. Exact mathematical formulations of the optimization problems
    3. Architecture details of the neural networks used
    4. Hyperparameters for both the models and optimization algorithms
    5. Data generation process and dataset statistics


[1] Cristian,  Rares,  et al.  "End-to-end learning for optimization via constraint-enforcing approximators."  AAAI 2023

**Questions:**

1. The paper would benefit greatly from a thorough revision to enhance clarity and readability. Please see my first comment in the weaknesses section.

2. Can you provide more theoretical grounds for your claims? Specifically around the issues of the gradients for the smoothed solvers. And why does your method resolve these issues?

3. Please provide more details on the experimental results. Again, please see my comments in the weakenesses section.

4. Can we see some experiments about the behavior of the gradients? Especially if you cannot provide theoretical results. For example, how does the magnitude of the gradients for your proposed approach compare with those of existing methods. Can we also see results on accuracy/regret as a function of the number of training epochs used.

5. Ideally, the same base model is used across all experiments and only the training method changes. That is, use the same base neural network and optimization solver for each dataset. Then, use each training method (smoothed, surrogate loss, your proposed combination). This way, everything is even across experiments for the same dataset, instead of using for instance both CVXPYlayers and DYS-Net etc.

---

> ### Author Response · Authors · 2024-11-21
> **Comment by Authors**
>
> - Regarding your comments that 'the empirical problem is not introduced', we explained the problem definition in the second paragrpah in Section 2. We explicitly mentioned there that an ML model is trained to map features to the parameter(s) and then a decision is made by solving the optimization problem with the predicted parameter(s). In the beginning of Section 3, we stated that the objective is to  minimize the empirical regret. Do you prefer it to be in more explicit, such as in Equation environment?
> - Thank you for suggesting to demonstrate the vanishing gradient phenomenon with computational experiments. Based on your suggestion, we have added computational experimentation on Top-K selection problems of multiple dimensions.
> In these experiments, we generate groundtruth and predicted cost by randomly sampling integers.
> We compute the solution to the optimization problem for groundtruth and predicted cost.
> Then we solve the `smoothed' optimization problem with predicted costs using CvxpyLayer, in order to compare the gradients of $reget$ and $\mathcal{L_{\textit{SCE}}}$.
> In Appendix B of the revised version, we have illustrated the zero gradient issue for different sizes of the Top-k selection problem.
> The experimental result supports the idea that **minimizing regret suffers from the zero gradient problem due to the plateau effect. However, the zero gradient problem does not appear
> when $\mathcal{L_{\textit{SCE}}}$ is minimized.**
> - We have explained in the text why regret results in a zero-gradient, whereas $\mathcal{L_{\textit{SCE}}}$ does not, by comparing Eq. (9) and Eq. (12). Due to the presence of the term $v^\star - v^\star (\hat{y})$, the gradient of $\mathcal{L_{\textit{SCE}}}$ does not reduce to zero (Line 290).
> - We used the same neural network architecture and same training and test datasets for all the techniques. Additionally, to evaluate the test instances, we consistently used the same solver—Gurobi, an exact combinatorial solver.
> The key difference between the techniques lies in the loss function and how the gradient of the loss function is computed.
>
> - We want to make some clarifications about the objective of the paper to help  you understand the setting.
> In order to train a model to minimize the empirical regret, we need the solution to the optimization problem with the predicted parameters. These optimization problems must be solved $n_{epoch} * n_{instances}$ times, which is computation-intensive.  Moreover, due to the combinatorial nature of the optimization problem, the gradient cannot backpropagate through the solver. Techniques under differentiable solver, instead considers a 'smoothed' version of the problem, which can be differentiated through to enable backpropagation and then minimize the empirical regret.
> Techniques under surrogate losses, minimizes a surrogate loss, instead of empirical regret.
> As explained in the text, the gradients of surrogate losses such as $\mathcal{L_{\textit{SPO}^+}}$ and $\mathcal{L_{\textit{SCE}}}$ depend on whether a combinatorial solver (which is non-differentiable) or a 'smoothed' differentiable solver is used. Note both can be used when a surrogate loss is minimized. However, when empirical regret is minimized we have to use a 'smoothed' differentiable solver. This is why $\mathcal{L_{\textit{SPO}^+}}$ and $\mathcal{L_{\textit{SCE}}}$ are listed twice in Table 3, but Regret only under CvxpyLayer.
> - CvxpyLayer solves the 'smoothed' problem optimally. DYS-Net, on the other hand, is a fast, neural-based approximate solver and can speed up training drastically. Because of a lack of guarantee that DYS-Net solves the problem optimally The research question is: can it result in lower regret on the test dataset?  We show that it is possible if the surrogate loss, $\mathcal{L_{\textit{SCE}}}$ is minimized during training. This has not been done in the literature so far.
>
> - Importantly, we want to clarify that for each technique, we use only one solver throughout the training process and do not mix solvers during training.
> - We had to introduce DYS-Net in the text, otherwise the readers will not understand the motivation behind it, which is it is a fast neural solver, capable of reducing training time in DFL.
> - Due to space constraints, we cannot provide detailed description of the experimental setups. As we used existing benchmarks from PyEpo (Tang & Khalil), we have referred that paper. Thank you for the suggestion to provide the details in the Appendix. We will also show regret as a function of the number of training epochs.
>
> 1. Tang, B., & Khalil, E. B. (2024). Pyepo: A pytorch-based end-to-end predict-then-optimize library for linear and integer programming. Mathematical Programming Computation, 16(3), 297-335.

---

### Official Review · Reviewer_HVPw · 2024-11-03

**Soundness:** 3
**Presentation:** 4
**Contribution:** 3
**Rating:** 8
**Confidence:** 4

**Summary:**

In the predict-then-optimize framework, this paper introduces an approach that combines smooth optimization, which transforms the solver into a smooth and differentiable form, with a surrogate loss that approximates regret. This combination allows the learning model to maintain meaningful gradient flow even in flat regions, enhancing stability during training. Additionally, the paper leverages DYS-Net, a fast differentiable solver, to significantly reduce training time while preserving decision quality.

**Strengths:**

1. **Novelty:** This paper first introduces a combination of smoothing techniques and a surrogate loss function within the predict-then-optimize framework for combinatorial optimization.
2. **Efficiency:** Solving the optimization problem is often the computational bottleneck in PtO. This paper demonstrates the advantages of using DYS-Net, a differentiable solver significantly reducing training time without compromising decision quality.
3. **Clarity:** The paper is well-organized and provides clear and thorough explanations. Additionally, visualizations and examples clarify the results and effectively support the theoretical insights.

**Weaknesses:**

1. **Lack of Theoretical Supports:** While the QP-based relaxation for ILP has shown promise in experimental results, the paper lacks a theory supporting the effectiveness of this relaxation approach. (I acknowledge that providing such theoretical support is challenging.)
2. **Lack of Comparison with Related Speed-Up Techniques:** While one of the main contributions of this paper is its focus on enhancing solver speed, it does not compare its approach with other established methods that also aim to accelerate predict-then-optimize (PtO) processes by leveraging ILP relaxations and caching strategies. For instance, 'Differentiation of Blackbox Combinatorial Solvers' and 'Pyepo: A PyTorch-Based End-to-End Predict-Then-Optimize Library for Linear and Integer Programming' utilize LP relaxations of ILP problems as an oracle to speed up solution times. Additionally, 'Contrastive Losses and Solution Caching for Predict-and-Optimize' introduces solution caching to improve efficiency. Comparing this paper’s approach with these alternative acceleration methods, rather than just focusing on ILP solvers, would provide a more comprehensive evaluation of the proposed method’s effectiveness in reducing computation time.
3. **Insufficient Scalability Demonstration:** Although the paper emphasizes computational efficiency, the experiments are conducted on relatively small instances (e.g., 11-node TSP). This may limit the understanding of the scalability for larger combinatorial problems.
4. **Missing Hyperparameter Details:** The paper does not provide explicit details on specific hyperparameters used for training, such as learning rate, batch size, or optimizer configurations. Thus, it is difficult for readers to reproduce the results. Most critically, there is a lack of information on the smoothing parameter $\mu$, which directly influences gradient flow and solution quality.

**Questions:**

1. Given that current exact solvers like Gurobi are highly efficient for solving QP problems, have you considered or conducted any comparisons between DYS-Net and other exact QP solvers in the forward pass?
2. Could the authors offer more insights or practical guidelines on choosing $\mu$?
3. Does DYS-Net need to be pre-trained before starting the main PtO training process? If so, could the authors provide some description and training time?
4. To improve reader comprehension, providing more details on DYS-Net would be highly beneficial.

---

> ### Author Response · Authors · 2024-11-21
> **Comment by Authors**
>
> - We appreciate the idea of comparing against the 'Solution Caching' approach. **We have added this experiment and presented the result in Figure 7 in the Appendix of the updated paper. We have found out DYS-Net has lower regret than solution caching**, although, solution caching is faster for the medium sized problem considered in the paper. However, DYS-Net scales much better when we consider larger optimization problems as shown in Figure 8 in the Appendix. This is because, the solution caching approach solves for small percentage of instances during training and for larger problems, even solving for small number of instances can pose a huge bottleneck.
> - In Figure 3, we indeed compared TSP instances till $11$ nodes.
> This is due to the fact that for larger TSP instances, we cannot complete training of $SPO_{+}^{combinatorial}$ and $SCE^{cvxpy}$.
> We conducted  more experiments on even larger TSP instances. We consider TSP instances with 15,20 and 25 nodes.
> In Figure 8 in the Appendix of the revised version, we compare DYS-Net (with $\mathcal{L_{\textit{SCE}}}$)  against solution caching on these larger TSP instances.
> - We will update the Appendix with more ablation analysis and table of hyperparameters.
>
> - $SPO_{+}^{combinatorial}$ and $SCE^{cvxpy}$ are trained using Gurobi MIP and cvxpylayer solver.
> $SPO_{+}^{combinatorial}$ solves the MIP problem, whereas cvxpylayer solves the 'smoothed' QP problem *till optimality*. It is clearly evident in Figure 3 that DYS-Net is much faster than these two.
> - We treat $\mu$ as a hyper parameter and choose the value of $\mu$ which minimizes regret on the validation dataset. We will further provide an ablation analysis on $\mu$.
> - No, it does not require any pre-training. We treat the $\alpha$ parameter in DYS-Net as hyperparameter.
> We directly adopted the approach of McKenzie, D., Heaton, H., & Fung, S. W. (2024). Nevertheless, we will add a Section in the Appendix to describe how we train DYS-Net.

---

> > ### Comment · Reviewer_HVPw · 2024-11-21
> >
> > We appreciate the thoughtful and comprehensive response from the authors to our comments and for conducting additional experiments to address the feedback provided. The effort to refine the manuscript and further substantiate the contributions is commendable. Below, I outline some key points and suggestions for further strengthening the paper:
> >
> > 1. **Emphasizing the Contribution to PtO Efficiency:** From my understanding, the major contribution of this paper is the improvement of the training efficiency of DFL through DYS-Net. Yes, computation for optimization is a significant computational bottleneck in combinatorial optimization problems. This contribution is valuable and deserves to be highlighted more prominently in the abstract and introduction. Ensuring that readers immediately recognize this aspect will help clarify the unique positioning of the work within the DFL literature.
> >
> > 2. **Additional Experiments with Solution Caching:** Including solution caching experiments in Appendix E is a strong addition and provides promising results. These experiments effectively claim the contribution of DYS-Net in scaling to larger optimization problems. Thank you for including this comparison.
> >
> > 3. **Scalability to Larger Problems:** Regarding scalability, it is worth noting that CaVE [1] has demonstrated experiments on larger instances, such as the TSP with 50 nodes and even CVRP problems. While DYS-Net shows promise in addressing scalability, additional experiments on larger-scale problems for SPO+ and PFY, similar to those in CaVE, may also be feasible. Exploring these could provide a more comprehensive scalability comparison and further highlight DYS-Net's advantages.
> >
> > 4. **CaVE as an Important Baseline:** Given the focus of this work on training efficiency, it is essential to consider CaVE [1] as an additional baseline. CaVE similarly claims to achieve both efficient training and state-of-the-art decision performance. While CaVE has limitations in its applicability to binary decision variables, this aligns closely with the benchmarks used in the paper, which also focus on binary linear programs.
> >
> > [1] CaVE: A Cone-Aligned Approach for Fast Predict-then-optimize with Binary Linear Programs.

---

> > > ### Author Response · Authors · 2024-11-22
> > >
> > > Thank you for acknowledging that we demonstrate improved training efficiency in DFL with DYS-Net while regret being comparable  to state-of-the-art methods like SPO and PFY. We will revise the abstract and introduction to better emphasize this contribution.
> > >
> > > We also agree that benchmarking against CaVE as a baseline is highly relevant. Thank you for this valuable feedback.

---

> ### Comment · Reviewer_HVPw · 2024-11-22
>
> I notice the potential of your method for application to larger-scale PtO. As you mentioned, training time in current PtO algorithms is often overheaded by the optimization solver, given that solving optimization problems tens of thousands of times is required. This bottleneck is indeed a critical practical limitation, and addressing it could significantly advance the field.
>
> However, to fully substantiate this contribution, the current set of experiments and benchmarks may not yet be sufficient. While I understand that the rebuttal time is limited, providing additional experimental results on larger problem scales and corresponding benchmarks would greatly strengthen the paper. If the authors can present more comprehensive results, I would be happy to adjust my score accordingly.

---

> > ### Author Response · Authors · 2024-11-28
> >
> > We want to thank you very much for suggesting this work. We have managed to compare the proposed approach of minimizing surrogate losses with DYS-Net against CaVE.
> > We acknowledge that CaVE introduces a novel and effective approach by minimizing the angular distance of the predicted cost vector from the optimality cone. This innovative idea allows it to bypass solving the optimization problem directly and demonstrates scalability in training time for the TSP problem instances.
> >
> > However, we find out significant limitations in CaVE's  generalization. For instance, in the shortest path problems, its runtime and memory consumption become prohibitively high due to the large number of active constraints.
> > or instance, in the TSP problems, the number of active constraints for 5, 10 and 15 node TSP are 20, 65 and 135.
> > However, for $5 \times 5$ ,  $10 \times 10$ ,  $15 \times 15$  shortest path grids the number of active constraints are 90, 380 and 3540 respectively.
> > In the knapsack problems,CaVE fails because it relies on storing only the active constraints; but for knapsack the capacity constraints are often inactive. So, CaVE approach misses out the critical constraints in such problems.
> >
> > These limitations of CaVE in such problems highlights the advantage of fast, scalable solvers that work across a diverse range of optimization problems.
> > This is why we work with DYS-Net, a scalable neural solver for (regularized) LPs.
> > In contrast to CaVE, our approach with DYS-Net minimizes surrogate losses, providing low regret with significantly reduced training time across different problems, including problems where CaVE either fails to scale or underperforms.

---

> > > ### Comment · Reviewer_HVPw · 2024-11-29
> > >
> > > Thank you for your detailed response and for addressing my comments thoroughly. I appreciate the insightful comparisons with CaVE and your innovative use of DYS-Net, particularly the relaxed LP approach for active constraints. This paper has made a significant contribution to decision-focused learning with clear advantages for scalability.
> > >
> > > As promised, I will increase my evaluation score to reflect the quality of your work. However, I encourage you to emphasize DYS-Net's scalability for larger problems more explicitly in experiments, as this is a critical strength that would further enhance the impact of your paper.
> > >
> > > Thank you again for your efforts, and good luck.

---

> > > > ### Author Response · Authors · 2024-11-29
> > > >
> > > > Thank you for your constructive suggestions and encouragement. Irrespective of the decision, we are also interested in testing DYS-Net's performances in other optimization problems like CVRP. Thanks a lot.

---

### Official Review · Reviewer_uyNm · 2024-11-04

**Soundness:** 2
**Presentation:** 2
**Contribution:** 2
**Rating:** 5
**Confidence:** 5

**Summary:**

The paper studies decision-focused learning (DFL) for tasks where the parameters appear linearly in the objective. Specifically, the paper proposes applying QP smoothing to existing decision-focused learning surrogate losses such as SPO+ and the self-contrastive estimation (SCE) loss. They motivate this application by pointing out that the typical approach of applying smoothing directly to the non-convex task loss may still have vanishing gradients, which makes optimizing the decision-focus loss challenging. In addition to smoothing, the paper also studies recently proposed approaches to making DFL more scalable. For both approaches, the paper provides numerical experiments demonstrating the efficacy of applying more heuristic approaches.

**Strengths:**

The paper highlights two popular decision-focused learning approaches, smoothing and surrogate losses, and points out reasonable drawbacks of learning with both approaches. It then shows how combining the two approaches may potentially help address weaknesses found in the two approaches. They also show how recently proposed approaches for making DFL more scalable can be combined with surrogate losses to improve scalability and the quality of the learned decisions. They then justify their claims with numerical experiments of a popular set of benchmarks found in PyEPO which highlights some robustness.

**Weaknesses:**

The paper's weaknesses can be broken down into two categories: i) issues related to theoretical justification and ii) issues with the numerical experiments.

Theoretical Issues
- The theoretical justification issues stem from the main argument of the paper--that smoothing task loss directly may not solve a zero-gradient issue which hampers gradient-based optimization techniques. The paper seems to claim that the benefit of combining smoothing and surrogate losses addresses this problem and produces losses with "better" gradients. However, the surrogate losses the paper proposes combining with smoothing are inherently convex, so it is unclear why smoothing would be beneficial. The authors could address this issue by analyzing the newly proposed surrogate loss directly. Perhaps they can verify whether the new surrogate loss formed by combining QP smoothing is convex or non-convex. In the latter case, it may make sense why "better" gradients may be beneficial for the surrogate loss. For the former, perhaps it suggests that QP smoothing adds some form of regularization instead providing better gradients. It may also be helpful to then compare the approach with combining surrogate losses directly with regularizers instead of replacing non-smooth components with smoothed components.

- On page 5 there is a proof, but no formal statement. Adding a formal statement that is proved as a lemma or proposition would improve the presentation and precision of the section.

Numerical Experiment Issues
- The overall more extensive numerical experiments could be performed. The recently accepted work to NeurIPS [1] has proposed a new smoothing approach that outperforms surrogates such as SPO+ and SCE for misspecified settings. Their numerics show this is true especially for settings where a zero regret policy can be learned. For example, in their experiments they propose a simple weighted classification problem and a shortest path problem where the optimal decision can always be learned from the context using a linear plug-in model. Including these experiments would be helpful since it could be implied that directly applying QP smoothing to the task loss would also perform well in these settings. This would further highlight the benefits or drawbacks of combining smoothing with the surrogate losses.

-The numerical experiments could also be expanded to include other methods such as the proposed surrogate loss in [1] and other approaches such as combining regularizers with the surrogate losses and the QP smoothed task loss. The latter approach would also help eliminate zero-gradient issues since the gradient of the regularizer may be non-zero almost everywhere.

# Update to Review
Based on the discussion it seemed the authors primarily decided to pursue answering Reviewer HVPw’s comments, so I decided to just follow along their discussion. I appreciate the additions as the authors clearly worked hard to produce the new results, which helped me better understand what the authors are trying to propose.

Based on these additions I still will keep my score the same. The primary reason is that the work still feels somewhat incremental to me. The authors seem to center their contribution around the SCE surrogate loss, with the main innovation being a combination of incorporating smoothing and DYS-Net into the surrogate loss. These additions don’t seem to greatly complicate the implementation of the SCE surrogate loss and thus feel like small upgrades. Additionally, these techniques are not novel by themselves, so it feels similar to heuristically adding a regularizer or a penalty term.

Below highlights more specific details:

1. From a computational point of view, the smoothed SCE loss with DYS-Net is indeed faster, which is expected. It performs comparably with other methods, but it is not uniformly or clearly the best in any setting. SPO+ with various computational tricks and CaVE do similar or better though not uniformly across all experiments.
2. Theoretically, it is hard to claim that SCE loss is more attractive as Proposition 1 only holds when the original problem Regret and SCE loss are zero. This is often not the case as seen in the experiments. Thus, it’s hard to rigorously argue that low SCE loss corresponds to low regret. This is evident in the generalization section of the appendix as the authors fail to show that the SCE loss bounds the expected regret.

The combination of these two points, lack of decisive numerics and limited theoretical justification, makes it hard for me to argue that the contribution is significant.

--------
[1] Huang, Michael, and Vishal Gupta. "Decision-Focused Learning with Directional Gradients." arXiv preprint arXiv:2402.03256 (2024).

**Questions:**

1. The proof on page 5 assumes that there exists choices of $\hat{\mathbf{y}}$ such that $\mathcal{L}\_{SCE}(\mathbf{v}^*(\hat{\mathbf{y}}), \mathbf{y})$ is 0 (such as $\hat{\mathbf{y}} = \mathbf{y}$). However, in most cases this is not true if your hypothesis class is misspecified or  if $\mathbf{y} = \mathbb{E}[\mathbf{y}] + \epsilon$ where $\epsilon$ is independent noise. Thus, how important is this result practically? Is often the case your surrogate loss equal or below 0? Also, in general, the surrogate loss is summed over samples, i.e. $\sum_{i=1}^n \mathcal{L}_{SCE}(\mathbf{v}^*(\hat{\mathbf{y}}_i), \mathbf{y}_i)$. It maybe beneficial to present the result relative to the empirical task-based loss.

2. What is the motivation for applying the QP smoothing to SPO+? SPO+ is already convex so it doesn't have any vanishing gradient issues. This seems to also be confirmed numerically since the QP smoothing does not provide significant performance gains when applied to SPO+

3. How does QP smoothing tune $\mu$ in the quadratic term? Presumably larger $\mu$ has more smoothing, but is more different from the solution of the original problem. Does this affect performance and how does affect the computation time?

4. Is there a reason DYS-net is only applied to SCE? Can DYS-net be applied to SPO+ as well or other surrogate losses that require solving an LP?

5. Is there a reason why you don't consider other approaches for eliminating zero gradients issues such as adding a regularizer directly to the surrogate loss or combining MSE loss with the surrogate loss? Both solutions would seem to produce "better" gradients.

---

> ### Author Response · Authors · 2024-11-21
> **Comment by Authors**
>
> - Based on your suggestion, we have now presented our claim in Page 5 with a formal Proposition environment.
> The proposition is: $\mathcal{L_{\textit{SCE}}}=0$ implies $regret=0$ and $regret=0$  implies $\mathcal{L_{\textit{SCE}}}=0$.
> The motivation behind this proposition is to justify readers why minimizing $\mathcal{L_{\textit{SCE}}}$ is a valid strategy in order to obtain low regret.
> Obviously, in practice, achieving $\mathcal{L_{\textit{SCE}}}=0$ is not feasible due to potential model mis-specifications or noisy observations. Therefore, we minimize $\mathcal{L_{\textit{SCE}}}$ as best as possible using gradient descent.
> We really appreciate the idea of presenting a theory on  empirical task-based loss. We will look into it.
> - You are absolutely right that $\mathcal{L_{\textit{SPO}^+} }$ is convex and there is no advantage of applying  QP smoothing to $\mathcal{L_{\textit{SPO}^+} }$ from theoretical point of view. However, $\mathcal{L_{\textit{SCE}}}$ is not convex. We illustrate it in  Appendix C with the same example optimization problem as Section 4. This is in line with the experimental findings that smoothing  $\mathcal{L_{\textit{SCE}}}=0$ leads to improvement in test regret, whether smoothing  $\mathcal{L_{\textit{SPO}^+} }$ does not.
> - We treat $\mu$ as a hyper parameter and choose the value of $\mu$ which minimizes regret on the validation dataset. We will further provide an ablation analysis on $\mu$.
> - You are right. DYS-Net can be trained to minimize  $\mathcal{L_{\textit{SPO}^+} }$.
> In the first experiment, we observe minimizing $\mathcal{L_{\textit{SPO}^+} }$ with a 'smoothed' solver does not result in low regret, but minimizing $\mathcal{L_{\textit{SCE}}}$ does. As DYS-Net is also a `smoothed' solver, we consider minimizing $\mathcal{L_{\textit{SCE}}}$ in Experiment 2.
> Moreover, we conducted experiments minimizing $\mathcal{L_{\textit{SPO}^+} }$ using DYS-Net. We report this result in Figure 6 in the Appendix of the modified version. You can see in most cases, minimizing $\mathcal{L_{\textit{SCE}}}$  results in lower regret than minimizing $\mathcal{L_{\textit{SPO}^+} }$ .
> - In another project, we tried combing combining MSE loss with surrogate loss such as $\mathcal{L_{\textit{SPO}^+} }$. However, the scale of $\mathcal{L_{\textit{SPO}^+} }$ is not comparable to MSE and hence finding a good balance between the two is not straightforward. We will experimentally support this in the coming days.
> Having said that, our primary motivation in this paper is to come up with a surrogate loss which is compatible with DYS-Net to speed up training. Regularization by combining with MSE would not reduce training time.

---

> > ### Comment · Reviewer_uyNm · 2024-11-21
> >
> > I appreciate the updates to the paper and the extra guidance highlighting the positioning of the paper!
> >
> > I do have some additional questions as some of the responses you provided feel very contradictory. Additionally, I have questions about the generality of the insights you are claiming the paper provides.
> >
> > 1. In your global response, you seem to claim that the DYS-Net authors didn't incorporate SPO+ into DYS-NET because "The authors of DYS-Net assumed that the true cost parameter  is not present in the training data; hence they cannot use the SPO+ technique". However, you see to have added a version of SPO+ that uses DYS-Net in the appendix. Has the setting changed between the original DYS-Net paper and this paper? Highlighting these differences may help with the clarity of the contribution of this paper.
> > 2. Can you provide a better description of how DYS-Net integrates into the surrogate losses? It seems like a general approach that all decision focused learning surrogates could benefit from computationally. Specifically it's unclear what the pseudo-algorithm looks like as the paper only really describes the projection idea, but not what the integrated version looks like. If you claim "Achieving low regret with significantly reduced training time should be considered the primary contribution of our paper, " I would like to know **how** and **why** it was achieved.
> > 3. Is there any special reason the SCE loss synergizes with DYS-Net? There are other surrogates, convex[1] and non-convex[2][3], that would benefit from not having to solve combinatorial optimization problems exactly.
> > 4. The TSP experiment looks kinda strange to me. Why are the results not monotone increase as you increase the number of nodes for SPO+, but monotone increasing for everybody else?
> > 5. Smoothness and the application of DYS-Net seem pretty orthogonal to me. Are you trying to say that for improving computation speed, we should try to consider surrogates that require solving for the combinatorial solution and thus use DYS-Net rather than differentiating solutions directly? I think this message is currently lost in the discussion and it would be helpful to highlight this fact more clearly.
> >
> > __________________
> > [1] Berthet, Quentin, et al. "Learning with differentiable pertubed optimizers." Advances in neural information processing systems 33 (2020): 9508-9519.
> >
> > [2] Pogančić, Marin Vlastelica, et al. "Differentiation of blackbox combinatorial solvers." International Conference on Learning Representations. 2020.
> >
> > [3] Huang, Michael, and Vishal Gupta. "Decision-Focused Learning with Directional Gradients." The Thirty-eighth Annual Conference on Neural Information Processing Systems.

---

> ### Author Response · Authors · 2024-11-23
>
> - In most predict-then-optimize problems, it is assumed that the true parameter, (denoted by $\mathcal{\mathbf{y} }$ ) is observed in the training data. Note that this is the case for most benchmarks in PyEPO. Moreover, in this case Regret can be computed, as its computation requires the presence of $\mathcal{\mathbf{y} }$.
> In the DYS-Net paper, the authors assume that $\mathcal{\mathbf{y}}$ is not available in the training data, making it impossible to compute regret directly. Consequently, surrogate losses such as $\mathcal{L_{\textit{SPO}^+}}$ or $\mathcal{L_{\textit{SCE}}}$ cannot be applied in that setting, leading them to adopt the squared decision error (SqDE) as the loss function.
>
> - However, the authors acknowledge that if $\mathcal{\mathbf{y}}$ is observed, $\mathcal{L_{\textit{SPO}^+}}$ becomes applicable. In this scenario, minimizing either SqDE or regret with DYS-Net yields worse regret compared to minimizing $\mathcal{L_{\textit{SPO}^+}}$ with a MIP solver. This is corroborated by our experimental results in Figure 3. Crucially, when minimizing $\mathcal{L_{\textit{SCE}}}$ with DYS-Net, we demonstrate that the regret is comparable to or even better than that achieved by minimizing $\mathcal{L_{\textit{SPO}^+}}$ with a MIP solver. This highlights the effectiveness of our approach in such settings.
>
> - We want to clarify that any loss functions can be minimized using DYS-Net. In Figure 6 of the updated Appendix, we present the results obtained when $\mathcal{L_{\textit{SPO}^+}}$ is used as the surrogate loss function using DYS-Net. The difference between $\mathcal{L_{\textit{SPO}^+}}$ and $\mathcal{L_{\textit{SCE}}}$ is that $\mathcal{L_{\textit{SCE}}}$ is not a convex function, as explained in Appendix C. So, using a smoothed solver is helpful for $\mathcal{L_{\textit{SCE}}}$, whereas $\mathcal{L_{\textit{SPO}^+}}$ is already a convex function. However, **$regret$ being zero does not imply $\mathcal{L_{\textit{SPO}^+}}= 0$.** You can see this in Figure 5 in the Appendix. In the interval $(0,2)$ regret is zero, but $\mathcal{L_{\textit{SPO}^+}}$ is not.
> On the other hand, we have demonstrated in Proposition 1 that $regret = 0$ implies $\mathcal{L_{\textit{SCE}}} = 0$ and vice versa. In this sense, minimizing $\mathcal{L_{\textit{SCE}}}$ is more directly connected to minimizing regret. This direct alignment is what we attribute to the better performance of $\mathcal{L_{\textit{SCE}}}$ compared to $\mathcal{L_{\textit{SPO}^+}}$ when we use DYS-Net.
>
> - We have added Algorithm3 to explain how $\mathcal{L_{\textit{SCE}}}$ is minimized using  a `smoothed' solver. In Algorithm 3, we can use DYS-Net to solve the smoothed optimization problem to make training faster.
>
> - We remark that $SPO_{+}^{combinatorial}$  solves the MIP problem, whereas DYS-Net and Cvxpy solve a relaxed problem. That is why the rise in runtime of $SPO_{+}^{combinatorial}$ is much more prominent.
>
> - We have now clarified that the two existing categories of approaches and our combined approach in Appendix I, supported by Algorithms 1, 2, and 3. Our main claim is that, instead of minimizing the regret of the 'smoothed' optimization problem (Algorithm 1), one should minimize the surrogate loss of the 'smoothed' optimization problem (Algorithm 3).
>  To improve scalability, we propose using DYS-Net to solve the 'smoothed' optimization problem. It is important to note that both $\mathcal{L_{\textit{SPO}^+}}$ and $\mathcal{L_{\textit{SCE}}}$ require solving an optimization problem, so DYS-Net can be applied with both surrogate loss functions and possibly with other surrogate losses too.

---

### Official Review · Reviewer_6pA7 · 2024-11-05

**Soundness:** 2
**Presentation:** 2
**Contribution:** 2
**Rating:** 3
**Confidence:** 3

**Summary:**

This paper studies decision-focused learning (DFL) for combinatorial optimization problems. The authors summarize the previous methods of DFL for combinatorial optimization.  To overcome the challenges, previous works either employ a differentiable surrogate loss or tuning the combinatorial optimization to a differentiable mapping. However, the authors point out that the derivative remains nearly zero for a large region by existing methods. To address this issue, the authors combine both approaches of smoothing the cost function and using a surrogate loss. The authors verified the performance of the proposed method by four DFL benchmarks.

**Strengths:**

The paper has the following strengths:
+ The paper provides a good summary on the existing works of DFL for combinatorial optimization.
+ The paper propose to combine the methods of smoothing the cost and using a surrogate loss and apply DYS-Net to solve the scalability issue.
+ Numerical results on common DFL benchmarks are given.

**Weaknesses:**

The paper can be improved in the following aspects.
- The paper argues that the derivative remains nearly zero for a lot of combinatorial problems but this argument is not verified. It would be better if the authors can give some illustrations based on some examples.
- The authors propose to combine the two existing methods of DFL for combinatorial optimization. However, the combination looks straightforward. Can the authors present if there are some technical challenges to combine the two methods?
- The authors apply DYS-Net in [D. McKenzie 2024] to address the scalability of DFL. However, it is unclear about the novelty about applying DYS-Net here. Is it just an application?
- The empirical results will be more convincing if more ablation empirical studies are given.

**Questions:**

Please see the questions in weakness.

---

> ### Author Response · Authors · 2024-11-21
> **Comment by Authors**
>
> - We actually provided a simple illustration in the paragraph 'A deep dive into the gradient landscape'. We illustrate  in Fig 2(a) for this one-dimensional problem, how **'smoothed regret' remains constant but non-zero for $\hat{y} < -\mu$ (We plot for $\mu =6$).**  It is quite straightforward to explain this behavior in this one-dimensional case.
>
> - This plateau effect would be true for large optimization problems too. However, it is difficult to visualize this for optimization problems with more number of parameters. However, to convince readers we have added computational experiments to highlight the zero gradient problem exists even in larger problems.
> In Appendix B of the revised version, we have illustrated the zero gradient due to plateau effect for different size of Top-k selection problem.
> - No, the implementation to combine these two paradigms involves almost no challenge and in fact straightforward.
> We adopted the DYS-Net proposed in McKenzie, D., Heaton, H., & Fung, S. W. (2024) and minimize $\mathcal{L_{\textit{SCE}}}$ instead of squared decision error loss, as done by them. The implementation is indeed easy and straightforward,  just plugging in a different loss function.
> Our paper shows minimizing the 'smoothed regret' is not the best choice because of the plateau effect. We advocate for using 'smoothed' surrogate loss functions.
> - The primary motivation behind using DYS-Net is to reduce training time, as it is a neural solver. As mentioned by the authors of the DYS-Net, minimizing the regret directly with DYS-Net **cannot attain regret as low as SPO+** (SPO+ using an LP solver). We show that minimizing **the surrogate loss using DYS-NET, we can achieve regret as low as SPO+**.  The reduction in runtime come from using DYS-Net.
> - Thanks for this suggestion of ablation analysis. In one week, We can run and report a result for different values of DYS-Net hyperparameter $\alpha$ and the smoothing parameter $\mu$. Let us know if we misunderstood your intentions or you meant ablation analysis on something else.

---

> ### Comment · Reviewer_6pA7 · 2024-11-22
>
> I appreciate the responses of the authors.
>
> I have a further question regarding the method.
> The authors claim that the contribution in terms of method is to combine the surrogate loss and smoothing of combinatorial optimization. The authors responded that the implementation to combine these two paradigms involves almost no challenge.  This is good. However, either the surrogate loss or the smoothing of combinatorial optimization is an existing method. A combination of two existing methods can be valuable, but it can be trivial in some cases. I hope to understand what new technical challenges the combination introduces in terms of the algorithm design.

---

> > ### Author Response · Authors · 2024-11-28
> >
> > The techniques under both 'surrogate loss' and 'smoothing of combinatorial optimization' have been well studied. However, prior work in DFL has the assumption that if a proxy differentiable layer can be designed for a combinatorial optimization problem, the best strategy is to minimize regret by differentiating through the proxy layer. Our paper challenges this foundational assumption and demonstrates that there is value in minimizing the surrogate loss functions even when optimization can be effectively proxied as a differentiable layer. Through empirical evidence, we show that minimizing surrogate losses often leads to lower regret than directly minimizing regret.
> > Our paper is not **centered on technical implementation but rather on demonstrating that minimizing surrogate losses can achieve better regret on test data than minimizing regret directly, when gradient-based training is used.** We show when we minimize the surrogate losses using DYS-Net we achieve lower regret on test instances.
> >
> > Regarding technical challenges, implementation of DYS-Net requires setting the values of some hyperaprameters, which we have discussed Appendix L in the revised version.

---

### Official Review · Reviewer_B7Fi · 2024-11-08

**Soundness:** 4
**Presentation:** 4
**Contribution:** 1
**Rating:** 5
**Confidence:** 2

**Summary:**

The paper concerns itself with combinatorial optimization (CO) problems, in the predict-then-optimize (PtO) regime.  The authors advocate that two main approaches are used to train ML models to minimize downstream CO error (Decision Focused Learning):

1. Minimizing a differentiable (convex) surrogate function, (instead of the regret).
2. Smoothing the problem to a PQ, and minimizing this proxy instead.

The authors argue that whilst the latter is differentiable, its gradient is zero almost everywhere, except at transition points. The paper suggests that minmizing a surrogate loss even after smoothing, and provides supporting empirical evidence (in the form of toy data sets) in addition to the presented theory.

**Strengths:**

The paper is well written, with a good structure and presents the mathematics in a clear and coherent way to the reader. The proposed minimization of a surrogate function (Sections 3 & 4) is well reinforced with the experiments section, and the motivations in Section 5 are pertinent for such methods to be deployed at scale in the future.

**Weaknesses:**

Whilst the paper is well written, and produces a coherent argument, I am reluctant to accept on contribution grounds. From what I can see, the majority of the paper is pulling together prior works, with limited novel contribution; (please note I am new to this exact line of research and could be missing something here, hence my low confidence score).

More precisely, a large portion of the paper is compiling prior problems / works, the surrogate losses in 3.2.1 and 3.2.2 are from existing works, and the experiments (whilst nice and supporting the argument), are toy-experiments and with little differing the cited literature. This should not detract from the papers clear strengths, and hence I politely suggest further contribution could be added either via i) further experimentation ii) novel theory / mathematics for surrogates  (or novel surrogates), which would render the paper acceptable to the conference upon resubmission.

**Questions:**

See suggested relevant literature that you may wish to cite in Section 3.1 (lines 136-147):


- [Berthet 2020] *Learning with Differentiable Perturbed Optimizers*.
- [Blondel 2020] *Fast differentiable sorting and ranking*
- [Jang 2016] *Categorical Reparameterization with Gumbel-Softmax*.
- [Peterson 2024] *Generalizing Stochastic Smoothing for Differentiation and Gradient Estimation*
- [Stewart 2023] *Differentiable Clustering with Perturbed Spanning Forests*.
- [Peterson 2024] *Differentiable Top-k Classification Learning*

From my limited understanding, it appears [Berthet 2020] addresses some of the problems discussed in this paper.

---

> ### Author Response · Authors · 2024-11-21
> **Official Comment by Authors**
>
> We acknowledge that our work builds upon existing tools and techniques from the DFL literature.
> However, prior works in differentiable optimization for DFL primarily focus on *directly minimizing regret through differentiable optimization layers*. In contrast, our work highlights the **overlooked benefit of minimizing surrogate loss using a smoothed differentiable layer.**
>
>  By minimizing $\mathcal{L_{\textit{SCE}}}$ with DYS-Net, we achieve regret comparable to state-of-the-art methods while significantly reducing training time and this opens up a paradigm shift in DFL literature.
> As we mentioned in the general comment, **achieving low regret with DYS-Net (with significantly low training time) should be considered the primary contribution of our paper.**
>
> In Table 3, we have actually compared against PFY, which has been proposed by Berthet 2020.
> We have mistakenly cited another paper. We have corrected it in the revised version.
> The other works are not directly related to our works, but we will mention them in related literature.
> 1. [Berthet 2020] Learning with Differentiable Perturbed Optimizers.

---

### Meta-Review · Area_Chair_Xm8h · 2024-12-19

**Metareview:**

This paper proposes a method for efficient decision-focused learning (DFL) in combinatorial optimization by combining smoothing techniques with surrogate losses. This approach aims to improve training scalability while maintaining competitive performance. Although the paper presents a clear and well-motivated solution with promising empirical results, it needs further development.  Specifically, it lacks strong theoretical justification, comprehensive comparisons with other speed-up techniques, and convincing evidence of scalability on larger problems.  Therefore, the paper requires further revisions to address these weaknesses before acceptance.

**Additional Comments On Reviewer Discussion:**

The reviewers all communicated with the authors during the discussion phase, but were not ultimately convinced.

---

### Decision · Program_Chairs · 2025-01-22

Reject